# A general framework of Riemannian adaptive optimization methods with a convergence analysis

**Hiroyuki Sakai**  *sakai0815@cs.meiji.ac.jp*
*Meiji University*

**Hideaki Iiduka**  *iiduka@cs.meiji.ac.jp*
*Meiji University*

**Reviewed on OpenReview:** *https://openreview.net/forum?id=knv4lQFVoE*

## Abstract

This paper proposes a general framework of Riemannian adaptive optimization methods. The framework encapsulates several stochastic optimization algorithms on Riemannian manifolds and incorporates the mini-batch strategy that is often used in deep learning. Within this framework, we also propose AMSGrad on embedded submanifolds of Euclidean space. Moreover, we give convergence analyses valid for both a constant and a diminishing step size. Our analyses also reveal the relationship between the convergence rate and mini-batch size. In numerical experiments, we applied the proposed algorithm to principal component analysis and the low-rank matrix completion problem, which can be considered to be Riemannian optimization problems. Python implementations of the methods used in the numerical experiments are available at `https://github.com/iiduka-researches/202408-adaptive`.

## 1 Introduction

Riemannian optimization (Absil et al., 2008; Sato, 2021) has received much attention in machine learning. For example, batch normalization (Cho & Lee, 2017), representation learning (Nickel & Kiela, 2017), and the low-rank matrix completion problem (Vandereycken, 2013; Cambier & Absil, 2016; Boumal & Absil, 2015) can be considered optimization problems on Riemannian manifolds. This paper focuses on Riemannian adaptive optimization algorithms for solving stochastic optimization problems on Riemannian manifolds. In particular, we treat Riemannian submanifolds of Euclidean space (e.g., unit spheres and the Stiefel manifold).

In Euclidean settings, adaptive optimization methods are widely used for training deep neural networks. There are many adaptive optimization methods, such as Adaptive gradient (AdaGrad) (Duchi et al., 2011), Adadelta (Zeiler, 2012), Root mean square propagation (RMSProp) (Hinton et al., 2012), Adaptive moment estimation (Adam) (Kingma & Ba, 2015), Yogi (Zaheer et al., 2018), Adaptive mean square gradient (AMSGrad) (Reddi et al., 2018), AdaFom (Chen et al., 2019), AdaBound (Luo et al., 2019), Adam with decoupled weight decay (AdamW) (Loshchilov & Hutter, 2019) and AdaBelief (Zhuang et al., 2020). Reddi et al. (2018) proposed a general framework of adaptive optimization methods that encapsulates many of the popular adaptive methods in Euclidean space.

Bonnabel (2013) proposed Riemannian stochastic gradient descent (RSGD), the most basic Riemannian stochastic optimization algorithm. In particular, Riemannian stochastic variance reduction algorithms, such as Riemannian stochastic variance-reduced gradient (RSVRG) (Zhang et al., 2016), Riemannian stochastic recursive gradient (RSRG) (Kasai et al., 2018), and Riemannian stochastic path-integrated differential estimator (R-SPIDER) (Zhang et al., 2018; Zhou et al., 2019), are based on variance reduction methods in Euclidean space. There are several prior studies on Riemannian adaptive optimization methods for specific Riemannian manifolds. In particular, Kasai et al. (2019) proposed a Riemannian adaptive stochastic gradient algorithm on matrix manifolds (RASA). RASA is an adaptive optimization method on matrix

manifolds (e.g., the Stiefel manifold or the Grassmann manifold), and gave a convergence analysis under the upper-Hessian bounded and retraction $L$-smooth assumptions (see (Kasai et al., 2019, Section 4) for details). However, RASA is not a direct extension of the adaptive optimization methods commonly used in deep learning, and it works only for diminishing step sizes. RAMSGrad (Bécigneul & Ganea, 2019) and modified RAMSGrad (Sakai & Iiduka, 2021), direct extensions of AMSGrad, have been proposed as methods that work on Cartesian products of Riemannian manifolds. In particular, Roy et al. (2018) proposed cRAMSProp and applied it to several Riemannian stochastic optimizations. However, they did not provide a convergence analysis of it. More recently, Riemannian stochastic optimization methods, sharpness-aware minimization on Riemannian manifolds (Riemannian SAM) (Yun & Yang, 2024) and Riemannian natural gradient descent (RNGD) (Hu et al., 2024), were proposed.

## 1.1 Motivations

Kasai et al. (2019) presented useful results indicating RASA with a diminishing step size converges to a stationary point of a continuously differentiable function defined on a matrix manifold. Meanwhile, other results have indicated that using constant step sizes makes the algorithm useful for training deep neural networks (Zaheer et al., 2018) and that the setting of the mini-batch size depends on the performance of the algorithms used to train deep neural networks (Smith et al., 2018; Sato & Iiduka, 2023). The main motivation of this paper is thus to gain an understanding of the theoretical performance relationship between Riemannian adaptive optimization methods using constant step sizes and ones using the mini-batch size. We are also interested in understanding the theoretical performance relationship between the Riemannian adaptive optimization methods using diminishing step sizes and those using the mini-batch size.

## 1.2 Contributions

Our first contribution is to propose a framework of adaptive optimization methods on Riemannian submanifolds of Euclidean space (Algorithm 1) that is based on the framework (Reddi et al., 2018, Algorithm 1) proposed by Reddi, Kale and Kumar for Euclidean space. A key challenge in designing adaptive gradient methods over Riemannian manifolds is the absence of a coordinate system like that in Euclidean space. In this work, we specifically focus on manifolds that are embedded in Euclidean space, a common setting in many applications. We address the issue by constructing algorithms that rely on projections onto the tangent space at each point on the manifold. Our framework incorporates the mini-batch strategy that is often used in deep learning. Important examples of Riemannian submanifolds of the Euclidean space include the unit sphere and the Stiefel manifold. Note that (Kasai et al., 2019) focuses exclusively on the RASA algorithm, which is defined only on matrix manifolds. In contrast, our framework can be defined on any manifold embedded in Euclidean space. Moreover, within this framework, we propose AMSGrad on embedded submanifolds of Euclidean space (Algorithm 2) as a direct extension of AMSGrad.

Our second contribution is to give convergence analyses of the proposed algorithm (Algorithm 1) by using a mini-batch size $b$ valid for both a constant step size (Theorem 3.5) and diminishing step size (Theorem 3.6). In particular, Theorem 3.5 indicates that, under some conditions, the sequence $(x_k)_{k=1}^\infty$ generated by the proposed algorithm with a constant step size $\alpha$ satisfies

$$\min_{k=1,\ldots,K} \mathbb{E}\left[\|\mathrm{grad}\, f(x_k)\|_2^2\right] \le \frac{1}{K}\sum_{k=1}^K \mathbb{E}\left[\|\mathrm{grad}\, f(x_k)\|_2^2\right] = \mathcal{O}\left(\frac{1}{K} + \frac{1}{b}\right).$$

We should note that Theorem 3.5 is an extension of the result in Zaheer et al. (2018, Corollary 2). Theorem 3.6 indicates that, under some conditions, the sequence $(x_k)_{k=1}^\infty$ generated by the proposed algorithm with a diminishing step size $\alpha_k$ satisfies

$$\min_{k=1,\ldots,K} \mathbb{E}\left[\|\mathrm{grad}\, f(x_k)\|_2^2\right] \le \frac{1}{K}\sum_{k=1}^K \mathbb{E}\left[\|\mathrm{grad}\, f(x_k)\|_2^2\right] = \mathcal{O}\left(\left(1 + \frac{1}{b}\right)\frac{\log K}{\sqrt{K}}\right).$$

Since the stochastic gradient (see (1)) using large mini-batch sizes is approximately the full gradient of $f$; intuitively, using large mini-batch sizes decreases the number of steps needed to find stationary points of

$f$. Theorems 3.5 and 3.6 indicate that, the larger the mini-batch size $b$ is, the smaller the upper bound of $\frac{1}{K}\sum_{k=1}^{K}\mathbb{E}[\|\operatorname{grad} f(x_k)\|_2^2]$ becomes and the fewer the required steps become. Hence, Theorems 3.5 and 3.6 would match our intuition.

Theorem 3.5 indicates that, although the proposed algorithm with a constant step size $\alpha$ and constant mini-batch size $b$ does not always converge, we can expect that converges with an increasing mini-batch size $b_k$ such that $b_k \le b_{k+1}$. In practice, it has been shown that increasing the mini-batch size (Byrd et al., 2012; Balles et al., 2017; De et al., 2017; Smith et al., 2018; Goyal et al., 2018) is useful for training deep neural networks with mini-batch optimizers. Motivated by Theorems 3.5 and 3.6, we give convergence analyses of the proposed algorithm by using an increasing mini-batch size $b_k$ valid for both a constant step size (Theorem 3.7) and diminishing step size (Theorem 3.8). Theorem 3.7 indicates that, under some conditions, the sequence $(x_k)_{k=1}^{\infty}$ generated by the proposed algorithm with a constant step size $\alpha$ and an increasing mini-batch size $b_k$ satisfies

$$\min_{k=1,\ldots,K} \mathbb{E}\left[\|\operatorname{grad} f(x_k)\|_2^2\right] \le \frac{1}{K}\sum_{k=1}^{K}\mathbb{E}\left[\|\operatorname{grad} f(x_k)\|_2^2\right] = \mathcal{O}\left(\frac{1}{K}\right).$$

That is, with increasing mini-batch sizes, it can find a stationary point of $f$, in contrast to Theorem 3.5. Moreover, Theorem 3.8 indicates that, under some conditions, the sequence $(x_k)_{k=1}^{\infty}$ generated by the proposed algorithm with a diminishing step size $\alpha_k$ and an increasing mini-batch size $b_k$ satisfies

$$\min_{k=1,\ldots,K} \mathbb{E}\left[\|\operatorname{grad} f(x_k)\|_2^2\right] \le \frac{1}{\sum_{k=1}^{K}\alpha_k}\sum_{k=1}^{K}\alpha_k \mathbb{E}\left[\|\operatorname{grad} f(x_k)\|_2^2\right] = \mathcal{O}\left(\frac{1}{\sqrt{K}}\right).$$

That is, with increasing mini-batch sizes, it has a better convergence rate than with constant mini-batch sizes (Theorem 3.6).

The third contribution is to numerically compare the performances of several methods based on Algorithm 1, including Algorithm 2, with the existing methods. In the numerical experiments, we applied the algorithms to principal component analysis (PCA) (Kasai et al., 2018; Roy et al., 2018) and the low-rank matrix completion (LRMC) problem (Boumal & Absil, 2015; Kasai et al., 2019; Hu et al., 2024), which can be considered to be Riemannian optimization problems. Based on our experiments, we can summarize the following key findings. For the PCA problem, RAMSGrad with a constant step size minimizes the objective function in fewer or comparable iterations compared with RSGD, RASA-L, RASA-R and RASA-LR, regardless of the dataset. For the LRMC problem, RAdam with a diminishing step size minimizes the objective function in fewer or about the same number of iterations compared with RSGD, RASA-L and RASA-R, irrespective of the dataset. As reported in Sakai & Iiduka (2021), RAMSGrad demonstrates robust performance regardless of the initial step size. For the PCA and LRMC problems (where LRMC adopts a diminishing step size), a grid search was conducted, allowing the selection of an initial step size that yielded even better results. This likely contributed to the strong performance of RAMSGrad across datasets. However, for the LRMC problem, RAMSGrad may underperform relative to RASA, depending on the dataset. It is worth noting that the objective function in the LRMC problem is complex, and assumptions such as retraction $L$-smoothness may not hold in this case. This could explain the less favorable results observed for RAMSGrad in this particular setting. Moreover, to support our theoretical analyses, we show that increasing the batch size decreases the number of steps needed to a certain index. In particular, Table 1 (resp. Table 2) supports the result shown in Theorem 3.5, demonstrating that the proposed algorithms with a constant (resp. diminishing) step size require fewer iterations to reduce the gradient norm below the threshold as the batch size increases. Tables 1 and 2 confirm that larger batch sizes lead to faster convergence in terms of the number of iterations, aligning with our theoretical analysis. Similar tables (Tables 11–16) for the remaining datasets and experiments can be found in Appendix E. Note that the choice of batch size plays a critical role in balancing the computational cost per iteration and the total number of iterations required for convergence. This is related to the concept of the critical batch size, as discussed in Sakai & Iiduka (2024), which minimizes a stochastic first-order oracle (SFO). However, estimating this optimal batch size is challenging due to the dependency on unknown parameters such as the gradient Lipschitz constant and the variance of the stochastic gradient. To address this challenge, an effective approach may involve dynamically increasing the batch size at regular intervals

during optimization, such as doubling or tripling it after a fixed number of iterations. This method has theoretical guarantees for convergence (see Theorems 3.7 and 3.8) and can adaptively approach an efficient balance between computational cost and convergence speed.

## 2 Mathematical preliminaries

Let $\mathbb{R}^d$ be a $d$-dimensional Euclidean space with inner product $\langle x, y \rangle_2 := x^\top y$, which induces the norm $\|\cdot\|_2$. Let $\mathbb{R}_{++}$ be the set of positive real numbers, i.e., $\mathbb{R}_{++} := \{x \in \mathbb{R} \mid x > 0\}$. $I_d$ denotes a $d \times d$ identity matrix. For square matrices $X, Y \in \mathbb{R}^{d \times d}$, we write $X \prec Y$ (resp. $X \preceq Y$) if $Y - X$ is a positive-definite (resp. positive-semidefinite) matrix. For two matrices $X$ and $Y$ of the same dimension, $X \odot Y$ denotes the Hadamard product, i.e., element-wise product. Let $\max(X, Y)$ be the element-wise maximum. Let $\mathcal{S}^d$ (resp. $\mathcal{S}_+^d$, $\mathcal{S}_{++}^d$) be the set of $d \times d$ symmetric (resp. symmetric positive-semidefinite, symmetric positive-definite) matrices, i.e., $S^d := \{X \in \mathbb{R}^{d \times d} \mid X^\top = X\}$, $\mathcal{S}_+^d := \{X \in \mathbb{R}^{d \times d} \mid X \succeq O\}$ and $\mathcal{S}_{++}^d := \{X \in \mathbb{R}^{d \times d} \mid X \succ O\}$. Let $\mathcal{D}^d$ be the set of $d \times d$ diagonal matrices. Let $\mathcal{O}_d$ be the orthogonal group, i.e., $\mathcal{O}_d := \{X \in \mathbb{R}^{d \times d} \mid X^\top X = I_d\}$. We denote the trace of a square matrix $A$ by $\mathrm{tr}(A)$. Let $\mathcal{O}$ be Landau's symbol; i.e., for a sequence $(x_k)_{k=0}^\infty$, we write $x_k = \mathcal{O}(g(k))$ as $k \to \infty$ if there exist a constant $C > 0$ and an index $k_0$ such that $|x_k| \le C|g(k)|$ for all $k \ge k_0$.

In this paper, we focus on Riemannian manifolds (Sakai, 1996) embedded in Euclidean space. Let $M$ be an embedded submanifold of $\mathbb{R}^d$. Moreover, let $T_x M$ be the tangent space at a point $x \in M$ and $TM$ be the tangent bundle of $M$. Let $0_x$ be the zero element of $T_x M$. The inner product $\langle \cdot, \cdot \rangle_2$ of a Euclidean space $\mathbb{R}^d$ induces a Riemannian metric $\langle \cdot, \cdot \rangle_x$ of $M$ at $x \in M$ according to $\langle \xi, \eta \rangle_x = \langle \xi, \eta \rangle_2 = \xi^\top \eta$ for $\xi, \eta \in T_x M \subset T_x \mathbb{R}^d \cong \mathbb{R}^d$. The norm of $\eta \in T_x M$ is defined as $\|\eta\|_x = \sqrt{\eta^\top \eta} = \|\eta\|_2$. Let $P_x : T_x \mathbb{R}^d \cong \mathbb{R}^d \to T_x M$ be the orthogonal projection onto $T_x M$ (see Absil et al. (2008)). For a smooth map $F : M \to N$ between two manifolds $M$ and $N$, $\mathrm{D}F(x) : T_x M \to T_{F(x)} N$ denotes the derivative of $F$ at $x \in M$. The Riemannian gradient $\mathrm{grad}\, f(x)$ of a smooth function $f : M \to \mathbb{R}$ at $x \in M$ is defined as a unique tangent vector at $x$ satisfying $\langle \mathrm{grad}\, f(x), \eta \rangle_x = \mathrm{D}f(x)[\eta]$ for any $\eta \in T_x M$.

**Definition 2.1** (Retraction). *Let $M$ be a manifold. Any smooth map $R : TM \to M$ is called a retraction on $M$ if it has the following properties.*

- *$R_x(0_x) = x$ for all $x \in M$;*

- *With the canonical identification $T_{0_x} T_x M \cong T_x M$, $\mathrm{D}R_x(0_x) = \mathrm{id}_{T_x M} : T_x M \to T_x M$ for all $x \in M$,*

*where $R_x$ denotes the restriction of $R$ to $T_x M$.*

### 2.1 Examples

The unit sphere $\mathbb{S}^{d-1} := \{x \in \mathbb{R}^d \mid \|x\|_2 = 1\}$ is an embedded manifold of $\mathbb{R}^d$. The tangent space $T_x \mathbb{S}^{n-1}$ at $x \in \mathbb{S}^{d-1}$ is given by $T_x \mathbb{S}^{n-1} = \{\eta \in \mathbb{R}^d \mid \eta^\top x = 0\}$. The induced Riemannian metric on $\mathbb{S}^{d-1}$ is given by $\langle \xi, \eta \rangle_x = \langle \xi, \eta \rangle_2 := \xi^\top \eta$ for $\xi, \eta \in T_x \mathbb{S}^{d-1}$. The orthogonal projection $P_x : \mathbb{R}^d \to T_x \mathbb{S}^{d-1}$ onto the tangent space $T_x \mathbb{S}^{n-1}$ is given by $P_x(\eta) = (I_d - xx^\top)\eta$ for $x \in \mathbb{S}^{d-1}$ and $\eta \in T_x \mathbb{S}^{d-1}$.

An important example is the Stiefel manifold (Absil et al., 2008, Chapter 3.3.2), which is defined as $\mathrm{St}(p, n) := \{X \in \mathbb{R}^{n \times p} \mid X^\top X = I_p\}$ for $n \ge p$. $\mathrm{St}(p, n)$ is an embedded manifold of $\mathbb{R}^{n \times d}$. The tangent space $T_x \mathrm{St}(p, n)$ at $X \in \mathrm{St}(p, n)$ is given by

$$T_X \mathrm{St}(p, n) = \{\eta \in \mathbb{R}^{n \times p} \mid X^\top \eta + \eta^\top X = O\}.$$

The induced Riemannian metric on $\mathrm{St}(p, n)$ is given by $\langle \xi, \eta \rangle_X = \mathrm{tr}(\xi^\top \eta)$ for $\xi, \eta \in T_X \mathrm{St}(p, n)$. The orthogonal projection onto the tangent space $T_X \mathrm{St}(p, n)$ is given by $P_X(\eta) = \eta - X \mathrm{sym}(X^\top \eta)$ for $X \in \mathrm{St}(p, n)$, $\eta \in T_X \mathrm{St}(p, n)$, where $\mathrm{sym}(A) := (A + A^\top)/2$. The Stiefel manifold $\mathrm{St}(p, n)$ reduces to the orthogonal groups when $n = p$, i.e., $\mathrm{St}(p, p) = \mathcal{O}_p$.

Moreover, we will also consider the Grassmann manifold (Absil et al., 2008, Chapter 3.4.4) $\mathrm{Gr}(p, n) := \mathrm{St}(p, n)/\mathcal{O}_p$. Let $X \in \mathrm{St}(p, n)$ be a representative of $[X] := \{XQ \mid Q \in \mathcal{O}_p\} \in \mathrm{Gr}(p, n)$. We denote the

horizontal lift of $\eta \in T_{[X]} \operatorname{Gr}(p, n)$ at $X$ by $\bar{\eta}_X \in T_X \operatorname{St}(p, n)$. The Riemannian metric of the Grassmann manifold $\operatorname{Gr}(p, n)$ is endowed with $\langle \xi, \eta \rangle_{[X]} := \langle \bar{\xi}_X, \bar{\eta}_X \rangle_2$ for $\xi, \eta \in T_{[X]} \operatorname{Gr}(p, n)$. The orthogonal projection onto the tangent space $T_{[X]} \operatorname{Gr}(p, n)$ is defined through

$$\overline{P_{[X]}(\eta)} = (I_n - XX^\top)\bar{\eta}_X,$$

for $[X] \in \operatorname{Gr}(p, n)$ and $\eta \in T_{[X]} \operatorname{Gr}(p, n)$. Note that while the Grassmann manifold itself is not an embedded submanifold, in practical implementations, computations involving it is often performed via the Stiefel manifold. This allows the proposed method to be applied. Such an approach was used in previous studies, including Kasai et al. (2019); Hu et al. (2024).

## 2.2 Riemannian stochastic optimization problem

We focus on minimizing an objective function $f : M \to \mathbb{R}$ of the form,

$$f(x) = \frac{1}{N} \sum_{i=1}^{N} f_i(x),$$

where $f_i$ is a smooth function for $i = 1, \ldots, N$. We use the mini-batch strategy as follows (see Iiduka (2024) for detail). $s_{k,i}$ is a random variable generated from the $i$-th sampling at the $k$-th iteration, and $\boldsymbol{s}_k := (s_{k,1}, \ldots, s_{k,b_k})^\top$ is independent of $(x_k)_{k=1}^\infty$, where $b_k (\leq N)$ is the batch size. To simplify the notation, we denote the expectation $\mathbb{E}_{\boldsymbol{s}_k}$ with respect to $\boldsymbol{s}_k$ by $\mathbb{E}_k$. From the independence of $\boldsymbol{s}_1, \boldsymbol{s}_2, \ldots, \boldsymbol{s}_k$, we can define the total expectation $\mathbb{E}$ by $\mathbb{E}_1 \mathbb{E}_2 \cdots \mathbb{E}_k$. We define the mini-batch stochastic gradient $\operatorname{grad} f_{B_k}(x_k)$ of $f$ at the $k$-th iteration by

$$\operatorname{grad} f_{B_k}(x_k) := \frac{1}{b_k} \sum_{i=1}^{b_k} \operatorname{grad} f_{s_{k,i}}(x_k). \tag{1}$$

Our main objective is to find a stationary point $x_\star \in M$ satisfying $\operatorname{grad} f(x_\star) = 0_{x_\star}$.

## 2.3 Proposed general framework of Riemannian adaptive methods

Reddi et al. (2018) provided a general framework of adaptive gradient methods in Euclidean space. We devised Algorithm 1 by generalizing that framework to an embedded manifold of $\mathbb{R}^d$. The main difference from the Euclidean setting is computing the projection of $H_k^{-1} m_k$ onto the tangent space $T_{x_k} M$ by the orthogonal projection $P_{x_k}$. Algorithm 1 requires sequences of maps, $(\phi_k)_{k=1}^\infty$ and $(\psi_k)_{k=1}^\infty$, such that $\phi_k : T_{x_1} M \times \cdots \times T_{x_k} M \to \mathbb{R}^d$ and $\psi_k : T_{x_1} M \times \cdots \times T_{x_k} M \to \mathcal{D}^d \cap \mathcal{S}_{++}^d$, respectively. Note that Algorithm 1 is still abstract because the maps $(\phi_k)_{k=1}^\infty$ and $(\psi_k)_{k=1}^\infty$ are not specified. Algorithm 1 is the extension of a general framework in Euclidean space proposed by Reddi, Kale and Kumar (Reddi et al., 2018). In the Euclidean setting (i.e., $M = \mathbb{R}^d$), the orthogonal projection $P_{x_k}$ yields an identity map and this corresponds to the Euclidean version of the general framework.

By setting the maps $(\phi_n)_{n=1}^\infty$ and $(\psi_n)_{n=1}^\infty$ in Algorithm 1 to those used by an adaptive optimization method in Euclidean space, any adaptive optimization method can be extended to Riemannian manifolds. In the following, we show some examples of extending the optimization algorithm in Euclidean space to Riemannian manifolds by using Algorithm 1.

Here, SGD is the most basic method; it uses

$$\phi_k(g_1, \ldots, g_k) = g_k, \quad \psi_k(g_1, \ldots, g_k) = I_d.$$

Algorithm 1 with these maps corresponds to RSGD (Bonnabel, 2013) in the Riemannian setting. AdaGrad (Duchi et al., 2011), the first adaptive gradient method in Euclidean space that propelled research on adaptive methods, uses the sequences of maps $\phi_k(g_1, \ldots, g_k) = g_k$ and

$$v_k = v_{k-1} + g_k \odot g_k,$$
$$\psi_k(g_1, \ldots, g_k) = \operatorname{diag}(\sqrt{v_{k,1}}, \ldots, \sqrt{v_{k,d}}) + \epsilon I_d,$$

---

**Algorithm 1** The general framework of Riemannian adaptive optimization methods on an embedded submanifold of $\mathbb{R}^d$.

---

**Require:** Initial point $x_1 \in M$, retraction $R : TM \to M$, step sizes $(\alpha_k)_{k=1}^\infty \subset \mathbb{R}_{++}$, sequences of maps $(\phi_k)_{k=1}^\infty$, $(\psi_k)_{k=1}^\infty$.
**Ensure:** Sequence $(x_k)_{k=1}^\infty \subset M$.

1: $k \leftarrow 1$.
2: **loop**
3:      $g_k = \operatorname{grad} f_{B_k}(x_k)$.
4:      $m_k = \phi_k(g_1, \ldots, g_k) \in \mathbb{R}^d$.
5:      $H_k = \psi_k(g_1, \ldots, g_k) \in \mathcal{D}^d \cap \mathcal{S}_{++}^d$.
6:      $x_{k+1} = R_{x_k}(-\alpha_k P_{x_k}(H_k^{-1} m_k))$.
7:      $k \leftarrow k + 1$.
8: **end loop**

---

where $v_0 = 0 \in \mathbb{R}^d$ and $\epsilon > 0$. Here, we will denote the $i$-th component of $v_k$ by $v_{k,i}$. The exponential moving average variant of AdaGrad is often used in deep-learning training. The basic variant is RMSProp (Hinton et al., 2012), which uses the sequences of maps $\phi_k(g_1, \ldots, g_k) = g_k$ and

$$v_k = \beta_2 v_{k-1} + (1 - \beta_2) g_k \odot g_k,$$
$$\psi_k(g_1, \ldots, g_k) = \operatorname{diag}(\sqrt{v_{k,1}}, \ldots, \sqrt{v_{k,d}}) + \epsilon I_d,$$

where $v_0 = 0 \in \mathbb{R}^d$ and $\epsilon > 0$. Both Algorithm 1 with these maps and cRMSProp (Roy et al., 2018) can be considered extensions of RMSProp to Riemannian manifolds. They differ from each other in that parallel transport is needed to compute the search direction of cRMSProp, but it is not needed in our method.

Adam (Kingma & Ba, 2015) is one of the most common variants; it uses the sequence of maps,

$$m_k = \beta_1 m_{k-1} + (1 - \beta_1) g_k, \quad \phi_k(g_1, \ldots, g_k) = \frac{m_k}{1 - \beta_1^{k+1}}, \tag{2}$$

and

$$v_k = \beta_2 v_{k-1} + (1 - \beta_2) g_k \odot g_k, \quad \hat{v}_k = \frac{v_k}{1 - \beta_2^{k+1}},$$
$$\psi_k(g_1, \ldots, g_k) = \operatorname{diag}(\sqrt{\hat{v}_{k,1}}, \ldots, \sqrt{\hat{v}_{k,d}}) + \epsilon I_d, \tag{3}$$

where $m_0 = 0 \in \mathbb{R}^d$ and $v_0 = 0 \in \mathbb{R}^d$. $\beta_1 = 0.9$, $\beta_2 = 0.999$ and $\epsilon = 10^{-8}$ are typically recommended values. Moreover, within the general framework (Algorithm 1), we propose the following algorithm as an extension of AMSGrad (Reddi et al., 2018) in Euclidean space.

## 3 Convergence analysis

### 3.1 Assumptions and useful lemmas

We make the following Assumptions 3.1 (A1)–(A4). (A1) and (A2) include the standard conditions. (A3) assumes the boundedness of the gradient. (A4) is an assumption on the Lipschitz continuity of the gradient. (A5) assumes that a lower bound exists.

**Assumption 3.1.** *Let $(x_k)_{k=1}^\infty$ be a sequence generated by Algorithm 1.*

*(A1)* $\mathbb{E}_k[\operatorname{grad} f_{s_{k,i}}(x_k)] = \operatorname{grad} f(x_k)$ *for all $k \geq 1$ and $i = 1, \ldots, b_k$.*

*(A2)* *There exists $\sigma^2 > 0$ such that*

$$\mathbb{E}_k\left[\left\|\operatorname{grad} f_{s_{k,i}}(x_k) - \operatorname{grad} f(x_k)\right\|_2^2\right] \leq \sigma^2,$$

*for all $k \geq 1$ and $i = 1, \ldots, b_k$.*

---

**Algorithm 2** AMSGrad on an embedded submanifold of $\mathbb{R}^d$.

---

**Require:** Initial point $x_1 \in M$, retraction $R : TM \to M$, step sizes $(\alpha_k)_{k=1}^\infty \subset \mathbb{R}_{++}$, mini-batch sizes $(b_k)_{k=1}^\infty \subset \mathbb{R}_{++}$ hyperparameters $\beta_1, \beta_2 \in [0, 1)$, $\epsilon > 0$.
**Ensure:** Sequence $(x_k)_{k=1}^\infty \subset M$.

1: Set $m_0 = 0$, $v_0 = 0$ and $\hat{v}_0 = 0$.
2: $k \leftarrow 1$.
3: **loop**
4:      $g_k = \operatorname{grad} f_{B_k}(x_k)$.
5:      $m_k = \beta_1 m_{k-1} + (1 - \beta_1)g_k$.
6:      $v_k = \beta_2 v_{k-1} + (1 - \beta_2)g_k \odot g_k$.
7:      $\hat{v}_k = \max(\hat{v}_{k-1}, v_k)$.
8:      $H_k = \operatorname{diag}(\sqrt{\hat{v}_{k,1}}, \ldots, \sqrt{\hat{v}_{k,d}}) + \epsilon I_d$.
9:      $x_{k+1} = R_{x_k}(-\alpha_k P_{x_k}(H_k^{-1}m_k))$.
10:     $k \leftarrow k + 1$.
11: **end loop**

---

*(A3) There exists $G, B > 0$ such that $\|\operatorname{grad} f(x_k)\|_2 \leq G$ and $\|\operatorname{grad} f_{B_k}(x_k)\|_2 \leq B$ for all $k \geq 1$.*

*(A4) There exists a constant $L > 0$ such that*

$$|\mathrm{D}(f \circ R_x)(\eta)[\eta] - \mathrm{D}f(x)[\eta]| \leq L \|\eta\|_2^2,$$

*for all $x \in M$, $\eta \in T_x M$.*

*(A5) $f$ is bounded below by $f_\star \in \mathbb{R}$.*

Assumptions (A1) and (A2) are satisfied when the probability distribution is uniform (see Appendix F) or when the full gradient is used, i.e., when the batch size equals the total number of data. (A1) and (A2) have been used to analyze adaptive methods in Euclidean space (Zaheer et al., 2018, Pages 3, 12, and 16) and on the Hadamard manifold (Sakai & Iiduka, 2024, (2.2) and (2.3)). Assumptions (A3)–(A5) are often used in both Euclidean spaces and Riemannian manifolds. Intuitively, in practical applications, these assumptions are satisfied because the search space of the algorithm is bounded. If $f$ is unbounded on an unbounded search space $X$, then (A5) does not hold. Although there is a possibility such that (A4) is satisfied, (A2) would not hold, even when the probability distribution is uniform. When $f$ is unbounded, there is no minimizer of $f$. Hence, (A5) is essential in order to analyze optimization methods. Moreover, (A4) is needed in order to use a retraction $L$-smooth (Proposition 3.2) that is an extension of the Euclidean-type descent lemma. (A3) holds when the manifold is compact (Absil et al., 2008) or through a slight modification of the objective function and the algorithm (Kasai et al., 2018).

It is known that if Assumption 3.1 (A4) holds, so does the following Proposition 3.2. This property is known as retraction $L$-smoothness (see Huang et al. (2015); Kasai et al. (2018) for details).

**Proposition 3.2.** *Suppose that Assumption 3.1 (A4) holds. Then,*

$$f(R_x(\eta)) \leq f(x) + \langle \operatorname{grad} f(x), \eta \rangle_2 + \frac{L}{2} \|\eta\|_2^2,$$

*for all $x \in M$ and $\eta \in T_x M$.*

### 3.2 Convergence analysis of Algorithm 1

The main difficulty in analyzing the convergence of adaptive gradient methods is due to the stochastic momentum $m_k = \phi_k(g_1, \ldots, g_k)$. As a way to overcome this challenge in Euclidean space, Zhou et al. (2024); Yan et al. (2018); Chen et al. (2019) defined a new sequence $z_k$,

$$z_k = x_k + \frac{\beta_1}{1 - \beta_1}(x_k - x_{k-1}).$$

However, since the embedded submanifold is not closed under addition in Euclidean space, this strategy does not work in the Riemannian setting. Therefore, by following the policy of Zaheer et al. (2018), let us analyze the case in which $\phi_k(g_1, \ldots, g_k) = g_k$. To simplify the notation, we denote the $i$-th component of $g_k$ (resp. $v_k, \hat{v}_k$) by $g_{k,i}$ (resp. $v_{k,i}, \hat{v}_{k,i}$).

**Lemma 3.3.** *Suppose that Assumption 3.1 (A4) holds. If $\phi_k(g_1, \ldots, g_k) = g_k$ and $H_k^{-1} \preceq \nu I_d$ for all $k \geq 1$ and some $\nu > 0$, then the sequence $(x_k)_{k=1}^{\infty} \subset M$ generated by Algorithm 1 satisfies*

$$f(x_{k+1}) \leq f(x_k) + \left\langle \operatorname{grad} f(x_k), -\alpha_k H_k^{-1} g_k \right\rangle_2 + \frac{L\alpha_k^2 \nu^2}{2} \|g_k\|_2^2,$$

*for all $k \geq 1$.*

*Proof.* See Appendix B. $\qquad\square$

**Theorem 3.4.** *Suppose that Assumptions 3.1 (A1)–(A5) hold. Moreover, let us assume that $\alpha_{k+1} \leq \alpha_k$, $\phi_k(g_1, \ldots, g_k) = g_k$, $\alpha_k H_k^{-1} \succeq \alpha_{k+1} H_{k+1}^{-1}$ and there exist $\mu, \nu > 0$ such that $\mu I_d \preceq H_k^{-1} \preceq \nu I_d$ for all $k \geq 1$. Then, the sequence $(x_k)_{k=1}^{\infty} \subset M$ generated by Algorithm 1 satisfies*

$$\sum_{k=1}^{K} \alpha_k \left( \mu - \frac{L\alpha_k \nu^2}{2} \right) \mathbb{E}\left[ \|\operatorname{grad} f(x_k)\|_2^2 \right] \leq C_1 + C_2 \sum_{k=1}^{K} \frac{\alpha_k^2}{b_k},$$

*for some constant $C_1, C_2 > 0$.*

**Remark:** Since Algorithm 2 satisfies $\hat{v}_{k+1,i} := \max(\hat{v}_{k,i}, v_{k+1,i}) \geq \hat{v}_{k,i}$, it together with $\alpha_{k+1} \leq \alpha_k$, leads to $\alpha_k/(\sqrt{\hat{v}_{k,i}} + \epsilon) \geq \alpha_{k+1}/(\sqrt{\hat{v}_{k+1,i}} + \epsilon)$. Moreover, from Lemma A.3,

$$\frac{1}{B + \epsilon} \leq \frac{1}{\sqrt{\hat{v}_{k,i}} + \epsilon} \leq \frac{1}{\epsilon},$$

which implies $(B + \epsilon)^{-1} I_d \preceq H_k^{-1} \preceq \epsilon^{-1} I_d$. Therefore, Algorithm 2 satisfies the assumption $\alpha_k H_k^{-1} \succeq \alpha_{k+1} H_{k+1}^{-1}$ and $\mu I_d \preceq H_k^{-1} \preceq \nu I_d$ with $\mu = (B + \epsilon)^{-1}$ and $\nu = \epsilon^{-1}$.

*Proof.* We denote $\operatorname{grad} f(x_k)$ by $g(x_k)$. First, let us consider the case of $k = 1$. From Lemma 3.3, we have

$$f(x_2) \leq f(x_1) + \left\langle g(x_1), -\alpha_1 H_1^{-1} g_1 \right\rangle_2 + \frac{L\alpha_1^2 \nu^2}{2} \|g_1\|_2^2.$$

By taking $\mathbb{E}_1[\cdot]$ of both sides, we obtain

$$\mathbb{E}_1[f(x_2)] \leq f(x_1) + \left\langle g(x_1), -\alpha_1 \mathbb{E}_1[H_1^{-1} g_1] \right\rangle_2 + \frac{L\alpha_1^2 \nu^2}{2} \mathbb{E}_1\left[ \|g_1\|_2^2 \right]$$

$$\leq f(x_1) + \left\langle g(x_1), -\alpha_1 \mathbb{E}_1[H_1^{-1} g_1] \right\rangle_2 + \frac{L\alpha_1^2 \nu^2}{2} \left( \frac{\sigma^2}{b_1} + \|g(x_1)\|_2^2 \right)$$

where the second inequality comes from Lemma A.2. By taking $\mathbb{E}[\cdot]$ of both sides and rearranging terms, we get

$$-\frac{L\alpha_1 \nu^2}{2} \mathbb{E}\left[ \|g(x_1)\|_2^2 \right] \leq f(x_1) - \mathbb{E}[f(x_2)] + \left\langle g(x_1), -\alpha_1 \mathbb{E}[H_1^{-1} g_1] \right\rangle_2 + \frac{L\alpha_1^2 \sigma^2 \nu^2}{2b_1}.$$

By adding $\alpha_1 \mu G^2$ to both sides, we obtain

$$\alpha_1 \mu G^2 - \frac{L\alpha_1 \nu^2}{2} \mathbb{E}\left[ \|g(x_1)\|_2^2 \right] \leq f(x_1) - \mathbb{E}[f(x_2)] + \frac{L\alpha_1^2 \sigma^2 \nu^2}{2b_1} + \underbrace{\left\langle g(x_1), -\alpha_1 \mathbb{E}[H_1^{-1} g_1] \right\rangle_2 + \alpha_1 \mu G^2}_{C_0}.$$

Here, we note that

$$\alpha_1 \mu \mathbb{E}\left[\|g(x_1)\|_2^2\right] \leq \alpha_1 \mu G^2.$$

Therefore, we have

$$\alpha_1 \left(\mu - \frac{L\alpha_1 \nu^2}{2}\right) \mathbb{E}\left[\|g(x_1)\|_2^2\right] \leq f(x_1) - \mathbb{E}[f(x_2)] + \frac{L\alpha_1^2 \sigma^2 \nu^2}{2b_1} + C_0. \tag{4}$$

Next, let us consider the case of $k \geq 2$. From Lemma 3.3, we have

$$f(x_{k+1}) \leq f(x_k) + \left\langle g(x_k), -\alpha_{k-1} H_{k-1}^{-1} g_k \right\rangle_2 + \left\langle g(x_k), (\alpha_{k-1} H_{k-1}^{-1} - \alpha_k H_k^{-1}) g_k \right\rangle_2 + \frac{L\alpha_k^2 \nu^2}{2} \|g_k\|_2^2$$

for all $k \geq 2$. From Assumption 3.1 (A3), Lemma C.1, and $\alpha_{k-1} H_{k-1}^{-1} - \alpha_k H_k^{-1} \succeq O$, we have

$$f(x_{k+1}) \leq f(x_k) + \left\langle g(x_k), -\alpha_{k-1} H_{k-1}^{-1} g_k \right\rangle_2 + GB \operatorname{tr}(\alpha_{k-1} H_{k-1}^{-1} - \alpha_k H_k^{-1}) + \frac{L\alpha_k^2 \nu^2}{2} \|g_k\|_2^2.$$

By taking $\mathbb{E}_k[\cdot]$ of both sides, we obtain

$$\begin{aligned}
\mathbb{E}_k[f(x_{k+1})] &\leq f(x_k) + \left\langle g(x_k), -\alpha_{k-1} H_{k-1}^{-1} \mathbb{E}_k[g_k] \right\rangle_2 \\
&\quad + GB \mathbb{E}_k[\operatorname{tr}(\alpha_{k-1} H_{k-1}^{-1} - \alpha_k H_k^{-1})] + \frac{L\alpha_k^2 \nu^2}{2} \mathbb{E}_k\left[\|g_k\|_2^2\right] \\
&\leq f(x_k) - \alpha_{k-1} \left\langle g(x_k), H_{k-1}^{-1} g(x_k) \right\rangle_2 \\
&\quad + GB \mathbb{E}_k[\operatorname{tr}(\alpha_{k-1} H_{k-1}^{-1} - \alpha_k H_k^{-1})] + \frac{L\alpha_k^2 \nu^2}{2} \left(\frac{\sigma^2}{b_k} + \|g(x_k)\|_2^2\right),
\end{aligned}$$

where the first inequality comes from the independence of $H_{k-1}^{-1}$ for $\boldsymbol{s}_k$ and the second inequality comes from Lemmas A.1 and A.2. Here, since $H_{k-1}^{-1} \succeq \mu I_d$ and $\alpha_k \leq \alpha_{k-1}$, it follows that

$$-\alpha_{k-1} \left\langle g(x_k), H_{k-1}^{-1} g(x_k) \right\rangle_2 \leq -\alpha_k \mu \|g(x_k)\|_2^2,$$

which implies

$$\mathbb{E}_k[f(x_{k+1})] \leq f(x_k) - \alpha_k \left(\mu - \frac{L\alpha_k \nu^2}{2}\right) \|g(x_k)\|_2^2 + GB \mathbb{E}_k[\operatorname{tr}(\alpha_{k-1} H_{k-1}^{-1} - \alpha_k H_k^{-1})] + \frac{L\alpha_k^2 \sigma^2 \nu^2}{2b_k}.$$

By taking $\mathbb{E}[\cdot]$ of both sides, we have

$$\mathbb{E}[f(x_{k+1})] \leq \mathbb{E}[f(x_k)] - \alpha_k \left(\mu - \frac{L\alpha_k \nu^2}{2}\right) \mathbb{E}\left[\|g(x_k)\|_2^2\right] + GB \mathbb{E}[\operatorname{tr}(\alpha_{k-1} H_{k-1}^{-1} - \alpha_k H_k^{-1})] + \frac{L\alpha_k^2 \sigma^2 \nu^2}{2b_k}.$$

Rearranging the above inequality gives us

$$\alpha_k \left(\mu - \frac{L\alpha_k \nu^2}{2}\right) \mathbb{E}\left[\|g(x_k)\|_2^2\right] \leq \mathbb{E}[f(x_k)] - \mathbb{E}[f(x_{k+1})] + GB \mathbb{E}[\operatorname{tr}(\alpha_{k-1} H_{k-1}^{-1} - \alpha_k H_k^{-1})] + \frac{L\alpha_k^2 \sigma^2 \nu^2}{2b_k}. \tag{5}$$

By summing (5) from $k = 2$ to $k = K$, we have

$$\begin{aligned}
\sum_{k=2}^{K} &\alpha_k \left(\mu - \frac{L\alpha_k \nu^2}{2}\right) \mathbb{E}\left[\|g(x_k)\|_2^2\right] \\
&\leq \mathbb{E}[f(x_2)] - \mathbb{E}[f(x_{K+1})] + GB \mathbb{E}[\operatorname{tr}(\alpha_1 H_1^{-1} - \alpha_K H_K^{-1})] + \sum_{k=2}^{K} \frac{L\alpha_k^2 \sigma^2 \nu^2}{2b_k}.
\end{aligned}$$

Since $\mu I_d \preceq H_K^{-1} \preceq \nu I_d$ for all $k \geq 1$, it follows that $\text{tr}(\alpha_1 H_1^{-1}) \leq \alpha_1 \nu d$ and $\text{tr}(\alpha_K H_K^{-1}) \geq 0$. Here, we note that $\mathbb{E}[f(x_{K+1})] \geq f_\star$, from Assumption 3.1 (A3). Therefore, we have

$$\sum_{k=2}^K \alpha_k \left( \mu - \frac{L\alpha_k \nu^2}{2} \right) \mathbb{E}\left[ \|g(x_k)\|_2^2 \right] \leq \mathbb{E}[f(x_2)] - f_\star + GB\alpha_1 \nu d + \sum_{k=2}^K \frac{L\alpha_k^2 \sigma^2 \nu^2}{2b_k}. \tag{6}$$

Here, by adding both sides of (4) and (6), we have

$$\sum_{k=1}^K \alpha_k \left( \mu - \frac{L\alpha_k \nu^2}{2} \right) \mathbb{E}\left[ \|g(x_k)\|_2^2 \right] \leq \underbrace{f(x_1) - f_\star + C_0 + GB\alpha_1 \nu d}_{C_1} + \underbrace{\frac{L\sigma^2 \nu^2}{2}}_{C_2} \sum_{k=1}^K \frac{\alpha_k^2}{b_k}.$$

This completes the proof. $\qquad \square$

Our convergence analysis (Theorem 3.4) allows the proposed framework (Algorithm 1) to use both constant and diminishing steps sizes. Theorems 3.5 and 3.6 are convergence analyses of Algorithm 1 with constant and diminishing steps sizes and a fixed mini-batch size, respectively.

**Theorem 3.5.** *Under the assumptions in Theorem 3.4 and assuming that the constant step size $\alpha_k := \alpha$ satisfies $0 < \alpha < 2\mu L^{-1}\nu^{-2}$ and the mini-batch size $b_k := b$ satisfies $0 < b \leq N$, the sequence $(x_k)_{k=1}^\infty \subset M$ generated by Algorithm 1 satisfies*

$$\frac{1}{K} \sum_{k=1}^K \mathbb{E}\left[ \|\text{grad } f(x_k)\|_2^2 \right] = \mathcal{O}\left( \frac{1}{K} + \frac{1}{b} \right).$$

*Proof.* We denote $\text{grad } f(x_k)$ by $g(x_k)$. From Theorem 3.4, we obtain

$$\frac{1}{K} \sum_{k=1}^K \alpha \left( \mu - \frac{L\alpha \nu^2}{2} \right) \mathbb{E}\left[ \|g(x_k)\|_2^2 \right] \leq \frac{C_1}{K} + \frac{C_2 \alpha^2}{b}. \tag{7}$$

Since $0 < \alpha < 2\mu L^{-1}\nu^{-2}$, it follows that $(2\alpha\mu - L\alpha^2 \nu^2)/2 > 0$. Therefore, dividing both sides of (7) by $(2\alpha\mu - L\alpha^2 \nu^2)/2$ gives

$$\frac{1}{K} \sum_{k=1}^K \mathbb{E}\left[ \|g(x_k)\|_2^2 \right] \leq \frac{2C_1}{2\alpha\mu - L\alpha^2 \nu^2} \cdot \frac{1}{K} + \frac{2C_2 \alpha^2}{2\alpha\mu - L\alpha^2 \nu^2} \cdot \frac{1}{b}.$$

This completes the proof. $\qquad \square$

**Theorem 3.6.** *Under the assumptions in Theorem 3.4 and assuming that the diminishing step size $\alpha_k := \alpha/\sqrt{k}$ satisfies $\alpha \in (0,1]$ and the mini-batch size $b_k := b$ satisfies $0 < b \leq N$, the sequence $(x_k)_{k=1}^\infty \subset M$ generated by Algorithm 1 satisfies*

$$\frac{1}{K} \sum_{k=1}^K \mathbb{E}\left[ \|\text{grad } f(x_k)\|_2^2 \right] = \mathcal{O}\left( \left( 1 + \frac{1}{b} \right) \frac{\log K}{\sqrt{K}} \right).$$

*Proof.* We denote $\text{grad } f(x_k)$ by $g(x_k)$. Since $(\alpha_k)_{k=1}^\infty$ satisfies $\alpha_k \to 0$ $(k \to \infty)$, there exists a natural number $k_0 \geq 1$ such that, for all $k \geq 1$, if $k \geq k_0$, then $0 < \alpha_k < 2\mu L^{-1}\nu^{-2}$. Therefore, we obtain

$$0 < \mu - \frac{L\alpha_k \nu^2}{2} < \mu,$$

for all $k \geq k_0$. From Theorem 3.4, we have

$$\sum_{k=k_0}^K \alpha_k \left( \mu - \frac{L\alpha_k \nu^2}{2} \right) \mathbb{E}\left[ \|g(x_k)\|_2^2 \right] \leq C_1 + \frac{C_2}{b} \sum_{k=1}^K \alpha_k^2 - \sum_{k=1}^{k_0-1} \alpha_k \left( \mu - \frac{L\alpha_k \nu^2}{2} \right) \mathbb{E}\left[ \|g(x_k)\|_2^2 \right],$$

for all $K \geq k_0$. Since $(\alpha_k)_{k=1}^{\infty}$ is monotone decreasing and $\alpha_k > 0$, we obtain

$$\alpha_K \left( \mu - \frac{L\alpha_{k_0}\nu^2}{2} \right) \sum_{k=k_0}^{K} \mathbb{E}\left[ \|g(x_k)\|_2^2 \right] \leq C_1 + \frac{C_2}{b} \sum_{k=1}^{K} \alpha_k^2 + \sum_{k=1}^{k_0-1} L\alpha_k^2 \nu^2 \mathbb{E}\left[ \|g(x_k)\|_2^2 \right].$$

Dividing both sides of this inequality by $2^{-1}K\alpha_K(2\mu - L\alpha_{k_0}\nu^2) > 0$ yields

$$\frac{1}{K} \sum_{k=k_0}^{K} \mathbb{E}\left[ \|g(x_k)\|_2^2 \right] \leq \frac{2}{K\alpha_K(2\mu - L\alpha_{k_0}\nu^2)} \left( C_1 + \frac{C_2}{b} \sum_{k=1}^{K} \alpha_k^2 + \sum_{k=1}^{k_0-1} L\alpha_k^2 \nu^2 \mathbb{E}\left[ \|g(x_k)\|_2^2 \right] \right)$$

$$= \frac{1}{K\alpha_K} \cdot \underbrace{\frac{2}{2\mu - L\alpha_{k_0}\nu^2} \left( C_1 + \sum_{k=1}^{k_0-1} L\alpha_k^2 \nu^2 \mathbb{E}\left[ \|g(x_k)\|_2^2 \right] \right)}_{C_3}$$

$$+ \frac{1}{bK\alpha_K} \cdot \underbrace{\frac{2C_2}{2\mu - L\alpha_{k_0}\nu^2}}_{C_4} \sum_{k=1}^{K} \alpha_k^2.$$

From this and $\alpha_K := \alpha/\sqrt{K} < 1$, we obtain

$$\frac{1}{K} \sum_{k=1}^{K} \mathbb{E}\left[ \|g(x_k)\|_2^2 \right] \leq \frac{1}{K\alpha_K} \left( C_3 + \frac{C_4}{b} \sum_{k=1}^{K} \alpha_k^2 \right) + \frac{1}{K\alpha_K} \sum_{k=1}^{k_0-1} \mathbb{E}\left[ \|g(x_k)\|_2^2 \right]$$

$$= \frac{1}{\alpha\sqrt{K}} \left( C_3 + \sum_{k=1}^{k_0-1} \mathbb{E}\left[ \|g(x_k)\|_2^2 \right] + \frac{C_4}{b} \sum_{k=1}^{K} \alpha_k^2 \right).$$

From $\alpha \in (0, 1]$, we have that

$$\sum_{k=1}^{K} \alpha_k^2 = \sum_{k=1}^{K} \frac{\alpha^2}{k} \leq \sum_{k=1}^{K} \frac{1}{k} \leq 1 + \int_1^K \frac{dt}{t} = 1 + \log K.$$

Therefore,

$$\frac{1}{K} \sum_{k=1}^{K} \mathbb{E}\left[ \|g(x_k)\|_2^2 \right] \leq \frac{1}{\alpha\sqrt{K}} \left( C_3 + \sum_{k=1}^{k_0-1} \mathbb{E}\left[ \|g(x_k)\|_2^2 \right] + \frac{C_4}{b} + \frac{C_4}{b} \log K \right).$$

This completes the proof. $\qquad\square$

Theorems 3.7 and 3.8 are convergence analyses of Algorithm 1 with constant and diminishing steps sizes and increasing mini-batch sizes, respectively.

**Theorem 3.7.** *Under the assumptions in Theorem 3.4 and assuming that the constant step size $\alpha_k := \alpha$ satisfies $0 < \alpha < 2\mu L^{-1}\nu^{-2}$ and the mini-batch size $b_k$ satisfies $0 < b_k \leq N$ and $\sum_{k=1}^{\infty} b_k^{-1} < \infty$, the sequence $(x_k)_{k=1}^{\infty} \subset M$ generated by Algorithm 1 satisfies*

$$\min_{k=1,\ldots,K} \mathbb{E}\left[ \|\operatorname{grad} f(x_k)\|_2^2 \right] \leq \frac{1}{K} \sum_{k=1}^{K} \mathbb{E}\left[ \|\operatorname{grad} f(x_k)\|_2^2 \right] = \mathcal{O}\left( \frac{1}{K} \right).$$

*Proof.* We denote $\operatorname{grad} f(x_k)$ by $g(x_k)$. From Theorem 3.4, we obtain

$$\frac{1}{K} \sum_{k=1}^{K} \alpha \left( \mu - \frac{L\alpha\nu^2}{2} \right) \mathbb{E}\left[ \|g(x_k)\|_2^2 \right] \leq \frac{C_1}{K} + \frac{C_2\alpha^2}{K} \sum_{k=1}^{K} \frac{1}{b_k}. \qquad (8)$$

Since $0 < \alpha < 2\mu L^{-1}\nu^{-2}$, it follows that $(2\alpha\mu - L\alpha^2\nu^2)/2 > 0$. Therefore, dividing both sides of (8) by $(2\alpha\mu - L\alpha^2\nu^2)/2$ gives

$$\frac{1}{K}\sum_{k=1}^{K}\mathbb{E}\left[\|g(x_k)\|_2^2\right] \leq \frac{2C_1}{2\alpha\mu - L\alpha^2\nu^2} \cdot \frac{1}{K} + \frac{2C_2\alpha^2}{2\alpha\mu - L\alpha^2\nu^2} \cdot \frac{1}{K}\sum_{k=1}^{K}\frac{1}{b_k}.$$

This completes the proof. □

An example of $b_k$ such that $\sum_{k=1}^{\infty} b_k^{-1} < \infty$ used in Theorem 3.7 is such that the mini-batch size is multiplied by $\delta > 1$ every step, i.e.,

$$b_t = \delta^t b_0. \tag{9}$$

For example, the mini-batch size defined by (9) with $\delta = 2$ makes batch size double each step. Moreover, the mini-batch size defined by (9) and the diminishing step size $\alpha_k := \alpha/\sqrt{k}$, where $\alpha \in (0,1]$, satisfy that $\sum_{k=1}^{\infty} \alpha_k^2 b_k^{-1} < \infty$ (see the assumption in Theorem 3.8).

**Theorem 3.8.** *Under the assumptions in Theorem 3.4 and assuming that the diminishing step size $\alpha_k := \alpha/\sqrt{k}$ satisfies $\alpha \in (0,1]$ and the mini-batch size $b_k$ satisfies $0 < b_k \leq N$ and $\sum_{k=1}^{\infty} \alpha_k^2 b_k^{-1} < \infty$, the sequence $(x_k)_{k=1}^{\infty} \subset M$ generated by Algorithm 1 satisfies*

$$\min_{k=1,\ldots,K}\mathbb{E}\left[\|\mathrm{grad}\,f(x_k)\|_2^2\right] \leq \frac{1}{\sum_{k=1}^{K}\alpha_k}\sum_{k=1}^{K}\alpha_k\mathbb{E}\left[\|\mathrm{grad}\,f(x_k)\|_2^2\right] = \mathcal{O}\left(\frac{1}{\sqrt{K}}\right).$$

*Proof.* We denote $\mathrm{grad}\,f(x_k)$ by $g(x_k)$. Since $(\alpha_k)_{k=1}^{\infty}$ satisfies $\alpha_k \to 0$ $(k \to \infty)$, there exists a natural number $k_0 \geq 1$ such that, for all $k \geq 1$, if $k \geq k_0$, then $0 < \alpha_k < 2\mu L^{-1}\nu^{-2}$. Therefore, we obtain

$$0 < \mu - \frac{L\alpha_k\nu^2}{2} < \mu,$$

for all $k \geq k_0$. From Theorem 3.4, we have

$$\sum_{k=k_0}^{K}\alpha_k\left(\mu - \frac{L\alpha_k\nu^2}{2}\right)\mathbb{E}\left[\|g(x_k)\|_2^2\right] \leq C_1 + C_2\sum_{k=1}^{K}\frac{\alpha_k^2}{b_k} - \sum_{k=1}^{k_0-1}\alpha_k\left(\mu - \frac{L\alpha_k\nu^2}{2}\right)\mathbb{E}\left[\|g(x_k)\|_2^2\right],$$

for all $K \geq k_0$. Since $(\alpha_k)_{k=1}^{\infty}$ is monotone decreasing and $\alpha_k > 0$, we obtain

$$\left(\mu - \frac{L\alpha_{k_0}\nu^2}{2}\right)\sum_{k=k_0}^{K}\alpha_k\mathbb{E}\left[\|g(x_k)\|_2^2\right] \leq C_1 + C_2\sum_{k=1}^{K}\frac{\alpha_k^2}{b_k} + \sum_{k=1}^{k_0-1}L\alpha_k^2\nu^2\mathbb{E}\left[\|g(x_k)\|_2^2\right].$$

Dividing both sides of this inequality by $2^{-1}(2\mu - L\alpha_{k_0}\nu^2) > 0$ yields

$$\sum_{k=k_0}^{K}\alpha_k\mathbb{E}\left[\|g(x_k)\|_2^2\right] \leq \frac{2}{2\mu - L\alpha_{k_0}\nu^2}\left(C_1 + C_2\sum_{k=1}^{K}\frac{\alpha_k^2}{b_k} + \sum_{k=1}^{k_0-1}L\alpha_k^2\nu^2\mathbb{E}\left[\|g(x_k)\|_2^2\right]\right)$$

$$= \underbrace{\frac{2}{2\mu - L\alpha_{k_0}\nu^2}\left(C_1 + \sum_{k=1}^{k_0-1}L\alpha_k^2\nu^2\mathbb{E}\left[\|g(x_k)\|_2^2\right]\right)}_{C_3}$$

$$+ \underbrace{\frac{2C_2}{2\mu - L\alpha_{k_0}\nu^2}}_{C_4}\sum_{k=1}^{K}\frac{\alpha_k^2}{b_k}.$$

From this and $\sum_{k=1}^{\infty} \alpha_k^2 {b_k}^{-1} < \infty$, we have that

$$\sum_{k=1}^{K} \alpha_k \mathbb{E}\left[\|g(x_k)\|_2^2\right] \leq \underbrace{C_3 + \sum_{k=1}^{k_0-1} \alpha_k \mathbb{E}\left[\|g(x_k)\|_2^2\right] + C_4 \sum_{k=1}^{K} \frac{\alpha_k^2}{b_k}}_{C_5},$$

which implies that

$$\frac{1}{\sum_{k=1}^{K} \alpha_k} \sum_{k=1}^{K} \alpha_k \mathbb{E}\left[\|g(x_k)\|_2^2\right] \leq \frac{C_5}{\sum_{k=1}^{K} \alpha_k}.$$

From $\alpha_k := \alpha/\sqrt{k}$, we obtain

$$\sum_{k=1}^{K} \alpha_k = \sum_{k=1}^{K} \frac{\alpha}{\sqrt{k}} \geq \alpha \int_1^{K+1} \frac{dt}{\sqrt{t}} = 2\alpha(\sqrt{K+1} - 1).$$

Therefore,

$$\frac{1}{\sum_{k=1}^{K} \alpha_k} \sum_{k=1}^{K} \alpha_k \mathbb{E}\left[\|g(x_k)\|_2^2\right] \leq \frac{C_5}{2\alpha(\sqrt{K+1} - 1)}.$$

This completes the proof. $\qquad\square$

## 4 Numerical experiments

We experimentally compared our general framework of Riemannian adaptive optimization methods (Algorithms 1) with several choices of $(\phi_n)_{n=1}^{\infty}$ and $(\psi_n)_{n=1}^{\infty}$ with the following algorithms:

- RSGD (Bonnabel, 2013): Algorithm 1 with $\phi_k(g_1, \ldots, g_k) = g_k$ and $\psi_k(g_1, \ldots, g_k) = I_d$.

- RASA-LR, RASA-L, RASA-R (Kasai et al., 2019, Algorithm 1): $\beta = 0.99$.

- RAdam: Algorithm 1 with $(\phi_n)_{n=1}^{\infty}$ defined by (2), $(\psi_n)_{n=1}^{\infty}$ defined by (3), $\beta_1 = 0.9$, $\beta_2 = 0.999$ and $\epsilon = 10^{-8}$.

- RAMSGrad: Algorithm 2 with $\beta_1 = 0.9$, $\beta_2 = 0.999$ and $\epsilon = 10^{-8}$.

We experimented with both constant and diminishing step sizes. For each algorithm, we searched in the set $\{10^{-1}, 10^{-2}, \ldots, 10^{-8}\}$ for the best initial step size $\alpha$ (both constant and diminishing). Note that the constant (resp. diminishing) step size was determined to be $\alpha_k = \alpha$ (resp. $\alpha_k = \alpha/\sqrt{k}$) for all $k \geq 1$. For each experiment, each data set was split into a training set and a test set. The experiments used a MacBook Air (M1, 2020) and macOS Monterey version 12.2 operating system. The algorithms were written in Python 3.12.1 with the NumPy 1.26.0 package and the Matplotlib 3.9.1 package. The Python implementations of the methods used in the numerical experiments are available at `https://github.com/iiduka-researches/202408-adaptive`.

### 4.1 Principal component analysis

We applied the algorithms to a principal component analysis (PCA) problem (Kasai et al., 2018; Roy et al., 2018). For $N$ given data points $x_1, \ldots, x_N \in \mathbb{R}^n$ and $p \ (\leq n)$, the PCA problem is equivalent to minimizing

$$f(U) := \frac{1}{N} \sum_{i=1}^{N} \left\|x_i - UU^\top x_i\right\|_2^2, \tag{10}$$

on the Stiefel manifold $\mathrm{St}(p, n)$. Therefore, the PCA problem can be considered to be optimization problem on the Stiefel manifold. In the experiments, we set $p$ to 10 and the batch size $b$ to $2^{10}$. We used the QR-based retraction on the Stiefel manifold $\mathrm{St}(p, n)$ (Absil et al., 2008, Example 4.1.3), which is defined by

$$R_X(\eta) := \mathrm{qf}(X + \eta),$$

for $X \in \mathrm{St}(p, n)$ and $\eta \in T_X \mathrm{St}(p, n)$, where $\mathrm{qf}(\cdot)$ returns the $Q$-factor of the QR decomposition.

We evaluated the algorithms on training images of the MNIST dataset (LeCun et al., 1998) and the COIL100 dataset (Nene et al., 1996). The MNIST dataset contains $28 \times 28$ gray-scale images of handwritten digits. It has a training set of 60,000 images and a test set of 10,000 images. We transformed every image into a 784-dimensional vector and normalized its pixel values to lie in the range of $[0, 1]$. Thus, we set $N = 60000$ and $n = 784$. The COIL100 dataset contains 7,200 normalized color camera images of the 100 objects taken from different angles. As in the previous study (Kasai et al., 2019), we resized them to $32 \times 32$ pixels. Moreover, we split them into an 80% training set and a 20% test set. Thus, we set $N = 5760$ and $n = 1024$.

Figure 1(a) (resp. Figure 1(b)) shows the performances of the algorithms with a constant (resp. diminishing) step size for the objective function values defined by (10) with respect to the number of iterations on the training set of the MNIST dataset, while Figures 2(a) and 2(b) present those on the test set. Moreover, Figures 5(a) and 5(b) show those on the training set of the COIL100 dataset, while Figures 6(a) and 6(b) present those on the test set. Figure 3(a) (resp. 3(b)) presents the performances of the algorithms with a constant (resp. diminishing) step size for the norm of the gradient of objective function defined by (10) with respect to the number of iterations on the training set of the MNIST dataset, while Figures 4(a) and 4(b) present those on the test set of the MNIST dataset. Moreover, Figures 7(a) and 7(b) show those on the training set of the COIL100 dataset, Figures 8(a) and 8(b) present those on the test set. The experiments were performed for three random initial points, and the thick line plots the average of all experiments. The area bounded by the maximum and minimum values is painted the same color as the corresponding line. Note that the upper part of Figures 3(a), 3(b), 4(a) and 4(b) are cut off. The initial values of the gradient norm in these figures are approximately 38, highlighting that the gradient norm was minimized during the optimization process. Furthermore, our results differ from those presented in Kasai et al. (2019) because of three key factors. In Kasai et al. (2019), the mini-batch size was fixed at 10, while in our experiments we used $2^{10}$. Furthermore, the scope of the grid search differs from that of Kasai et al. (2019). Moreover, we experimented with a doubly increasing batch size every 100 steps from an initial batch size of $b_1 = 2^7$ and constant and diminishing step sizes. The results are given in Appendix G.

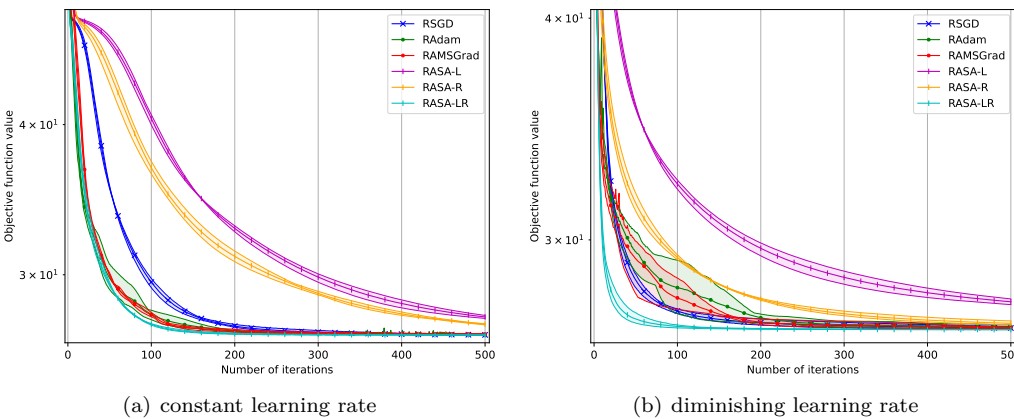

(a) constant learning rate          (b) diminishing learning rate

Figure 1: Objective function value defined by (10) versus number of iterations on the training set of the MNIST datasets.

Figures 1(a) and 2(a) indicate that RAdam and RAMSGrad (Algorithm 2) performed comparably to RASA-LR in the sense of minimizing the objective function value. Figures 1(b) and 2(b) indicate that RAdam and RAMSGrad (Algorithm 2) outperformed RASA-L and RASA-R. Figures 3(a) and 4(a) show that RAMSGrad (Algorithm 2) performed better than RASA-LR in the sense of minimizing the full gradient norm of the

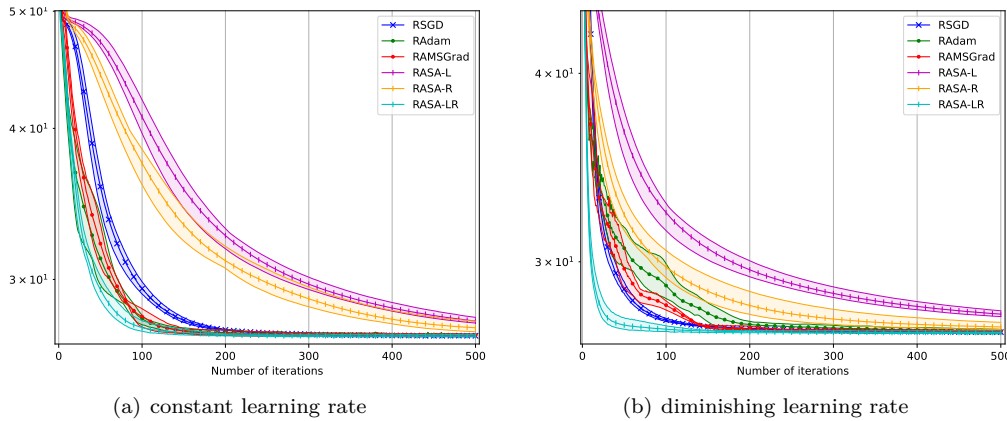

(a) constant learning rate

(b) diminishing learning rate

Figure 2: Objective function value defined by (10) versus number of iterations on the test set of the MNIST datasets.

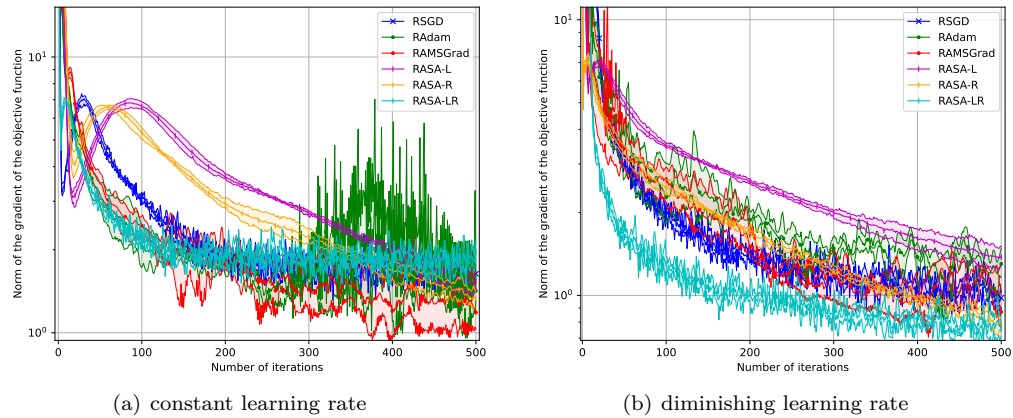

(a) constant learning rate

(b) diminishing learning rate

Figure 3: Norm of the gradient of objective function defined by (10) versus number of iterations on the training set of the MNIST datasets.

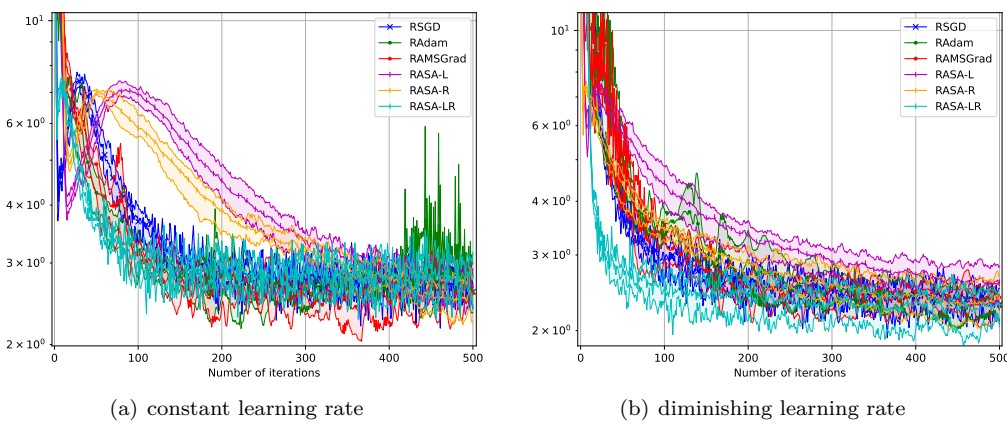

(a) constant learning rate

(b) diminishing learning rate

Figure 4: Norm of the gradient of objective function defined by (10) versus number of iterations on the test set of the MNIST datasets.

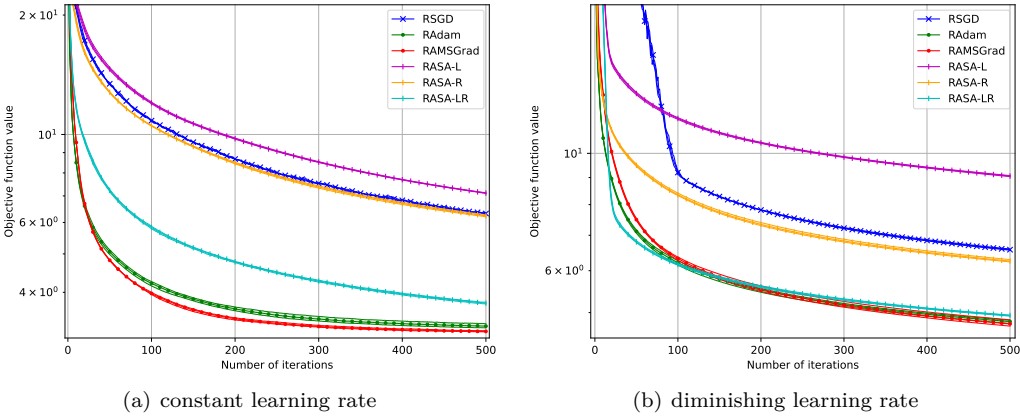

(a) constant learning rate

(b) diminishing learning rate

Figure 5: Objective function value defined by (10) versus number of iterations on the training set of the COIL100 datasets.

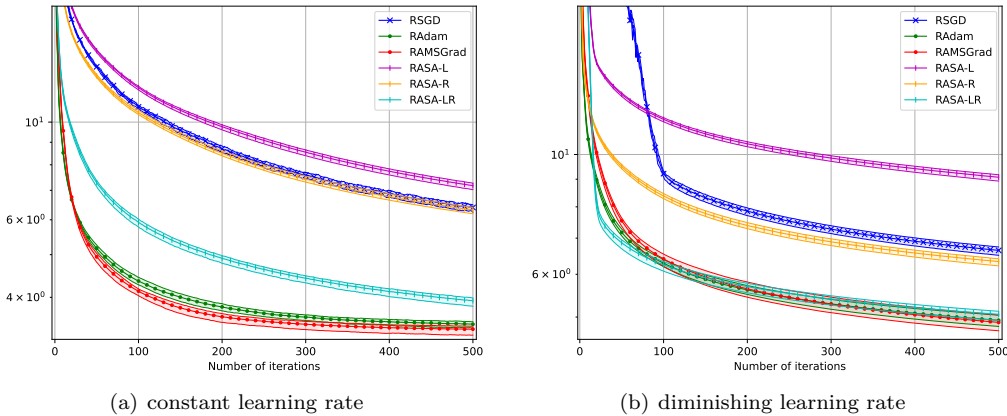

(a) constant learning rate

(b) diminishing learning rate

Figure 6: Objective function value defined by (10) versus number of iterations on the test set of the COIL100 datasets.

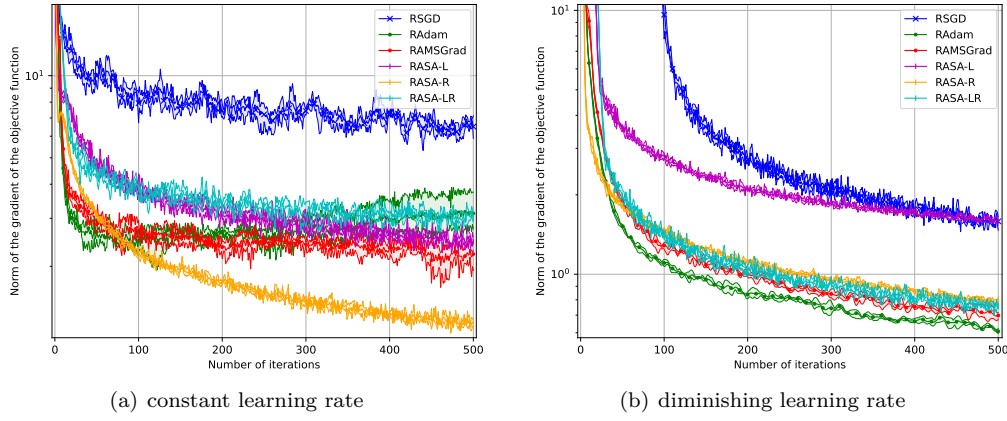

(a) constant learning rate

(b) diminishing learning rate

Figure 7: Norm of the gradient of objective function defined by (10) versus number of iterations on the training set of the COIL100 datasets.

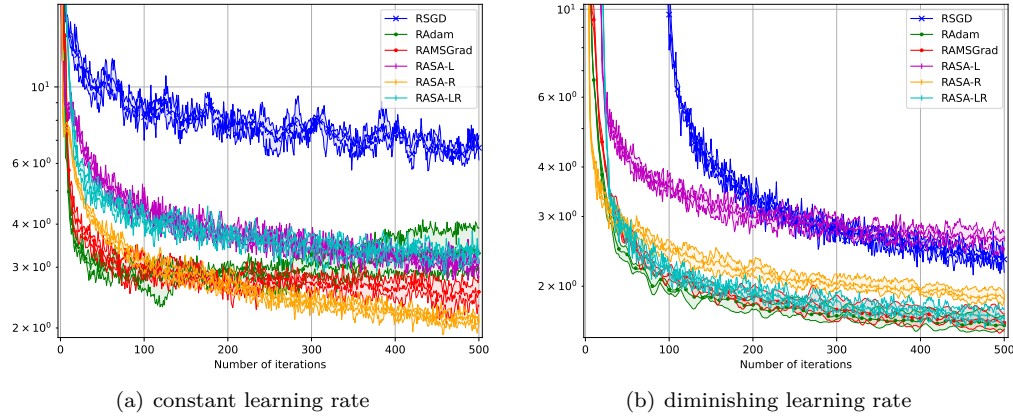

(a) constant learning rate                      (b) diminishing learning rate

Figure 8: Norm of the gradient of objective function defined by (10) versus number of iterations on the test set of the COIL100 datasets.

objective function. Figures 5(a) and 6(a) indicate that RAdam and RAMSGrad (Algorithm 2) had the best performance in the sense of minimizing the objective function value. Figures 5(b) and 6(b) indicate that RAdam and RAMSGrad (Algorithm 2) performed comparably to RASA-LR. Figures 7(a) and 8(a) shows that RAdam had the best performance in the sense of minimizing the full gradient norm of the objective function. Figures 7(b) and 8(b) indicate that RAdam and RAMSGrad (Algorithm 2) performed comparably to RASA-R and RASA-LR.

Table 1 compares performances across different batch sizes, showing the number of iterations and CPU time (in seconds) required by each algorithm with a constant step size to reduce the gradient norm below 2 when solving the PCA problem on the MNIST dataset. As our analysis (Theorem 3.5) indicates, increasing the batch size reduces the number of iterations needed to decrease the norm.

Table 1: Iterations and CPU time (seconds) required by each algorithm with a constant step size to reduce the gradient norm below 2 for solving the PCA problem on the MNIST dataset across different batch sizes. "-" indicates cases where the algorithm did not reach the threshold within the maximum allowed (1,000) iterations.

|  | batch size $b$ | number of iterations | CPU time (seconds) |
| --- | --- | --- | --- |
| RSGD | 256 | - | - |
|  | 512 | 303 | 0.531 |
|  | 1024 | 149 | 0.394 |
| RAdam | 256 | 391 | 0.532 |
|  | 512 | 150 | 0.270 |
|  | 1024 | 126 | 0.349 |
| RAMSGrad | 256 | 190 | 0.257 |
|  | 512 | 140 | 0.251 |
|  | 1024 | 114 | 0.323 |
| RASA-L | 256 | 493 | 1.228 |
|  | 512 | 442 | 1.291 |
|  | 1024 | 373 | 1.476 |
| RASA-R | 256 | 397 | 0.662 |
|  | 512 | 371 | 0.809 |
|  | 1024 | 300 | 0.901 |
| RASA-LR | 256 | - | - |
|  | 512 | 289 | 0.862 |
|  | 1024 | 113 | 0.447 |

Table 2 compares performances across different batch sizes, showing the number of iterations and CPU time (in seconds) required by each algorithm with a diminishing step size to reduce the gradient norm below 2 when solving the PCA problem on the MNIST dataset. As our analysis (Theorem 3.6) indicates, increasing the batch size reduces the number of iterations needed to decrease the norm. Similar tables (Tables 11–16) for the remaining datasets and experiments can be found in Appendix E.

Table 2: Iterations and CPU time (seconds) required by each algorithm with a diminishing step size to reduce the gradient norm below 2 for solving the PCA problem on the MNIST dataset across different batch sizes.

|  | batch size $b$ | number of iterations | CPU time (seconds) |
|---|---|---|---|
| RSGD | 256 | 239 | 0.315 |
|  | 512 | 140 | 0.327 |
|  | 1024 | 85 | 0.229 |
| RAdam | 256 | 292 | 0.434 |
|  | 512 | 189 | 0.349 |
|  | 1024 | 101 | 0.280 |
| RAMSGrad | 256 | 234 | 0.324 |
|  | 512 | 224 | 0.437 |
|  | 1024 | 141 | 0.384 |
| RASA-L | 256 | 409 | 1.008 |
|  | 512 | 370 | 1.133 |
|  | 1024 | 282 | 1.102 |
| RASA-R | 256 | 235 | 0.399 |
|  | 512 | 194 | 0.418 |
|  | 1024 | 156 | 0.472 |
| RASA-LR | 256 | 92 | 0.229 |
|  | 512 | 63 | 0.192 |
|  | 1024 | 36 | 0.145 |

## 4.2 Low-rank matrix completion

We applied the algorithms to the low-rank matrix completion (LRMC) problem (Boumal & Absil, 2015; Kasai et al., 2019; Hu et al., 2024). The LRMC problem aims to recover a low-rank matrix from an incomplete matrix $X = (X_{ij}) \in \mathbb{R}^{n \times N}$. We denote the set of observed entries by $\Omega \subset \{1, \ldots, n\} \times \{1, \ldots, N\}$, i.e., $(i, j) \in \Omega$ if and only if $X_{ij}$ is known. Here, we defined the orthogonal projection $P_{\Omega_i} : \mathbb{R}^n \to \mathbb{R}^n : a \mapsto P_{\Omega_i}(a)$ such that the $j$-th element of $P_{\Omega_i}(a)$ is $a_j$ if $(i, j) \in \Omega$, and 0 otherwise. Moreover, we defined $q_i : \mathbb{R}^{n \times p} \times \mathbb{R}^n \to \mathbb{R}^p$ as

$$q_i(U, x) := \operatorname*{arg\,min}_{a \in \mathbb{R}^p} \| P_{\Omega_i}(Ua - x) \|_2 , \tag{11}$$

for $i \geq 1$. By partitioning $X = (x_1, \ldots, x_N)$, the rank-$p$ LRMC problem is equivalent to minimizing

$$f(U) := \frac{1}{2N} \sum_{i=1}^{N} \| P_{\Omega_i}(Uq_i(U, x_i) - x_i) \|_2^2 , \tag{12}$$

on the Grassmann manifold $\mathrm{Gr}(p, n)$. Therefore, the rank-$p$ LRMC problem can be considered to be an optimization problem on the Grassmann manifold (see Hu et al. (2024, Section 1) or Kasai et al. (2019, Section 6.3) for details).

We evaluated the algorithms on the MovieLens-1M[1] datasets (Harper & Konstan, 2015) and the Jester[2] datasets for recommender systems. The MovieLens-1M datasets contains 1,000,209 ratings given by 6,040

---

[1] https://grouplens.org/datasets/movielens/
[2] https://grouplens.org/datasets/jester

users on 3,952 movies. Moreover, we split them into an 80% training set and a 20% test set. Thus, we set $N = 4832$ and $n = 3952$. The Jester dataset contains ratings of 100 jokes given by 24,983 users with scores from $-10$ to $10$. In addition, we split them into an 80% training set and a 20% test set. Thus, we set $N = 19986$ and $n = 100$.

In the experiments, we set $p$ to 10 and the batch size $b$ to $2^8$. We used `numpy.linalg.lstsq`[3] to solve the least squares problem (11). We used a retraction based on a polar decomposition on the Grassmann manifold $\mathrm{Gr}(p, n)$ (Absil et al., 2008, Example 4.1.3), which is defined through

$$\overline{R_{[X]}(\eta)} := (X + \bar{\eta}_X)(I_p + \bar{\eta}_X^\top \bar{\eta}_X)^{-\frac{1}{2}},$$

for $[X] \in \mathrm{Gr}(p, n)$ and $\eta \in T_{[X]} \mathrm{Gr}(p, n)$.

Figure 9(a) (resp. Figure 9(b)) shows the performances of the algorithms with a constant (resp. diminishing) step size for objective function values defined by (12) with respect to the number of iterations on the training set of the MovieLens-1M dataset, while Figures 10(a) and 10(b) present those on the Jester dataset. Moreover, Figures 13(a) and 13(b) show those on the training set of the Jester dataset, while Figures 14(a) and 14(b) present those on the test set of the Jester dataset. Figure 11(a) (resp. 11(b)) shows the performances of the algorithms with a constant (resp. diminishing) step size for the norm of the gradient of the objective function defined by (10) with respect to the number of iterations on the training set of the MovieLens-1M dataset, while Figures 12(a) and 12(b) present those on the test set of the MovieLens-1M dataset. Moreover, Figures 15(a) and 15(b) show those on the training set of the Jester dataset, while Figures 16(a) and 16(b) present those on the test set of the Jester dataset. The experiments were performed for three random initial points, and the thick line plots the average results of all experiments. The area bounded by the maximum and minimum values is painted the same color as the corresponding line. Moreover, we experimented with a doubly increasing batch size every 20 steps from an initial batch size of $b_1 = 2^6$ and constant and diminishing step sizes. The results are given in Appendix G.

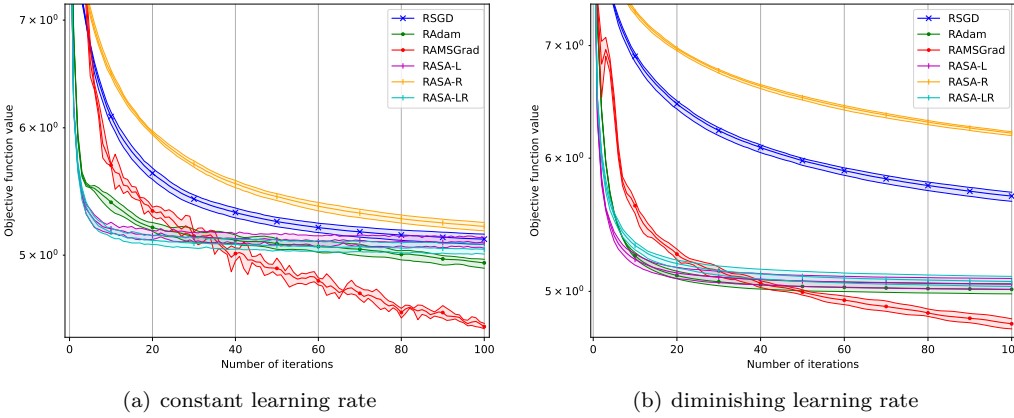

(a) constant learning rate          (b) diminishing learning rate

Figure 9: Objective function value defined by (12) versus number of iterations on the training set of the MovieLens-1M datasets.

Figures 9(a) and 10(a) indicate that RAMSGrad (Algorithm 2) performed better than RASA-L and RASA-LR in the sense of minimizing the objective function value. Figures 9(b) and 10(b) show that RAdam and RAMSGRad (Algorithm 2) performed comparably to RASA-L and RASA-LR. Figures 11(a) and 12(a) indicate that RAdam performed comparably to RASA-R in the sense of minimizing the full gradient norm of the objective function. Figures 11(b) and 12(b) show that RAdam outperformed RASA-R. Figures 13(a), 13(b), 14(a) and 14(b) indicate that RAMSGrad (Algorithm 2) performed better than RASA-LR in the sense of minimizing the objective function value. Figures 15(a) and 16(a) indicate that RAdam performed comparably to RASA-L and RASA-R in the sense of minimizing the full gradient norm of the objective function. Figures 15(b) and 16(b) show that RAMSGrad (Algorithm 2) outperformed RASA-L and RASA-R.

---

[3]https://numpy.org/doc/1.26/reference/generated/numpy.linalg.lstsq.html

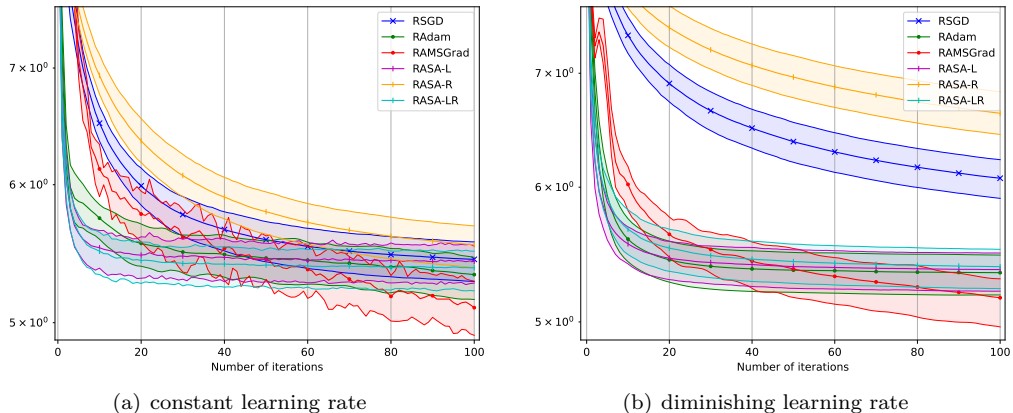

(a) constant learning rate

(b) diminishing learning rate

Figure 10: Objective function value defined by (12) versus number of iterations on the test set of the MovieLens-1M datasets.

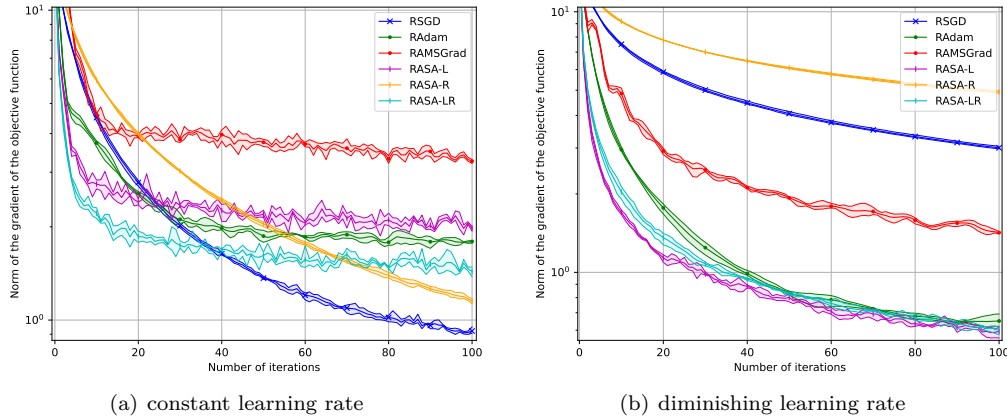

(a) constant learning rate

(b) diminishing learning rate

Figure 11: Norm of the gradient of objective function defined by (12) versus number of iterations on the training set of the MovieLens-1M datasets.

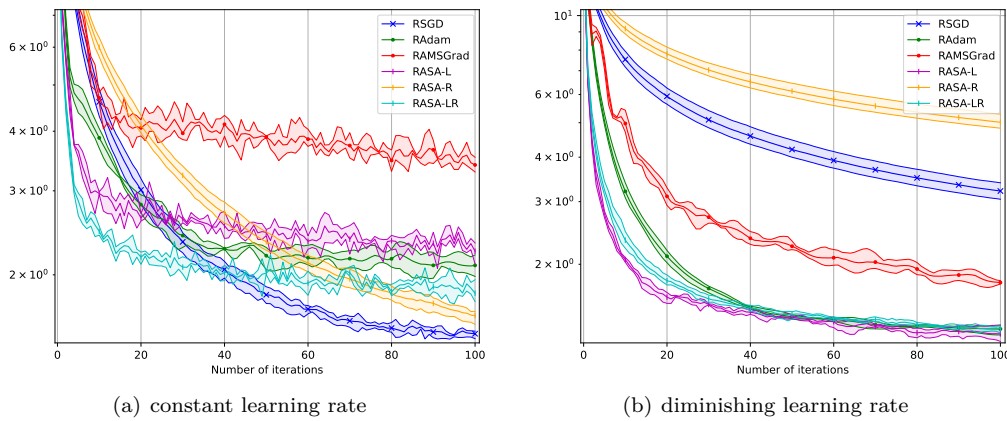

(a) constant learning rate

(b) diminishing learning rate

Figure 12: Norm of the gradient of objective function defined by (12) versus number of iterations on the test set of the MovieLens-1M datasets.

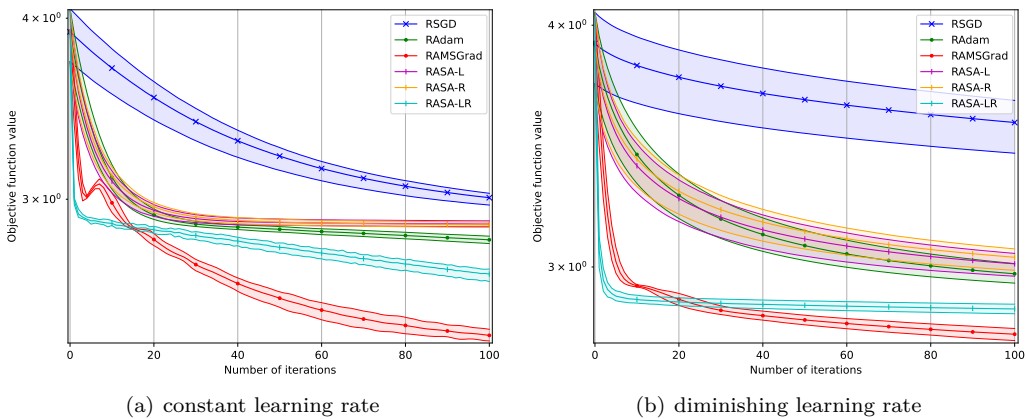

(a) constant learning rate

(b) diminishing learning rate

Figure 13: Objective function value defined by (12) versus number of iterations on the training set of the Jester datasets.

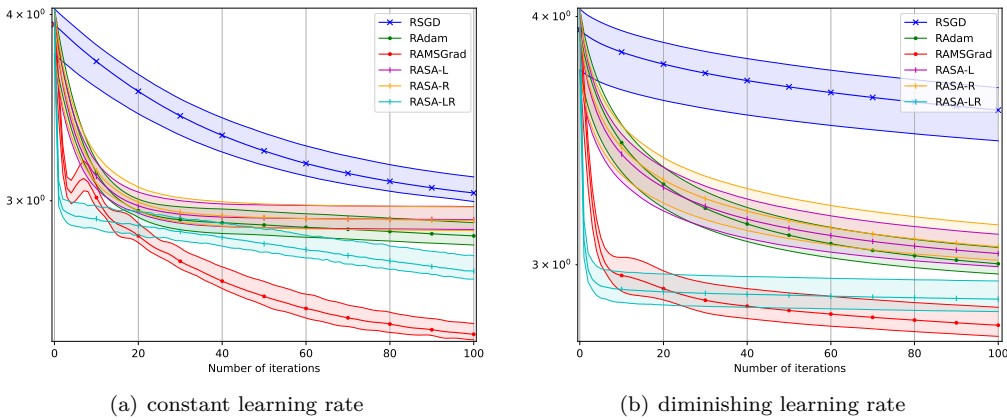

(a) constant learning rate

(b) diminishing learning rate

Figure 14: Objective function value defined by (12) versus number of iterations on the test set of the Jester datasets.

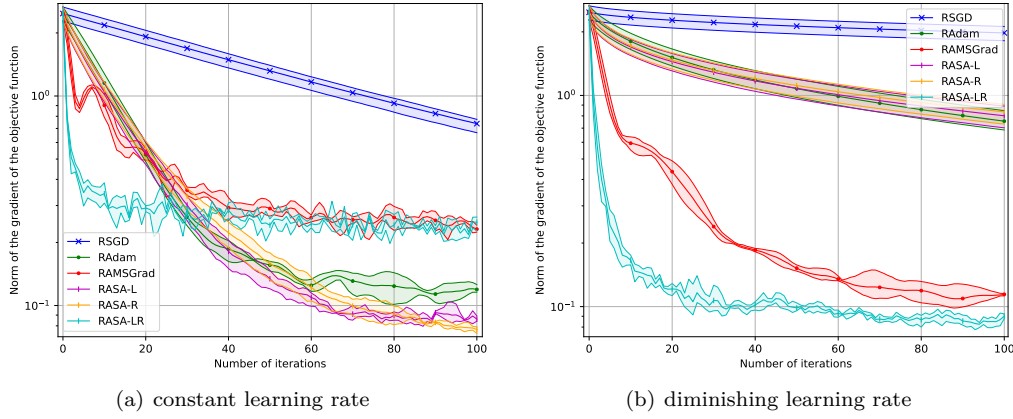

(a) constant learning rate

(b) diminishing learning rate

Figure 15: Norm of the gradient of objective function defined by (12) versus number of iterations on the training set of the Jester datasets.

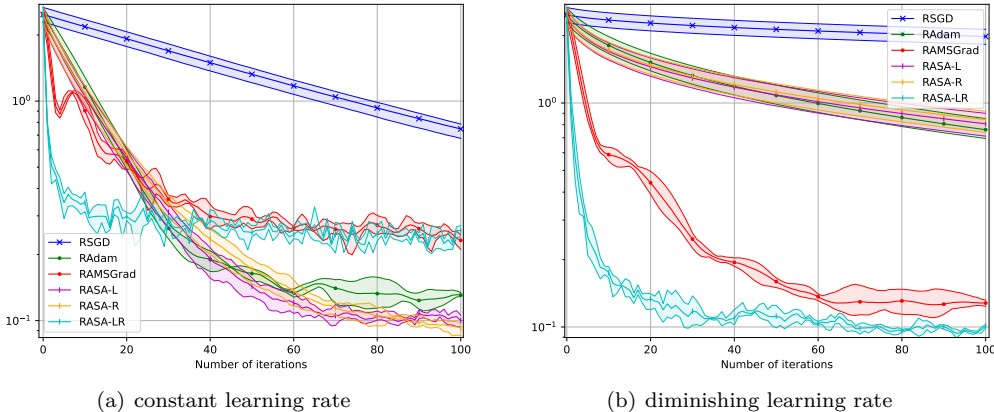

(a) constant learning rate  (b) diminishing learning rate

Figure 16: Norm of the gradient of objective function defined by (12) versus number of iterations on the test set of the Jester datasets.

## 5 Conclusion

This paper proposed a general framework of Riemannian adaptive optimization methods, which encapsulates several stochastic optimization algorithms on Riemannian manifolds. The framework incorporates the mini-batch strategy often used in deep learning. We also proposed RAMSGrad (Algorithm 2) that works on embedded submanifolds in Euclidean space within our framework. In addition, we gave convergence analyses that are valid for both a constant and diminishing step size. The analyses also revealed the relationship between the convergence rate and mini-batch size. We numerically compared the RAMSGrad (Algorithm 2) with the existing algorithms by applying them to principal component analysis and low-rank matrix completion problems, which can be considered to be Riemannian optimization problems. Numerical experiments showed that the proposed method performs well against PCA. RAdam and RAMSGrad performed well for constant and diminishing step sizes especially on the COIL100 dataset.

### Acknowledgments

We are sincerely grateful to the Action Editor, Stephen Becker, and the three anonymous reviewers for helping us improve the original manuscript. This work was supported by a JSPS KAKENHI Grant, number JP23KJ2003.

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

## A   Useful Lemmas

**Lemma A.1.** *Suppose that Assumption 3.1 (A1) holds. Let $(x_k)_{k=1}^\infty$ be a sequence generated by Algorithm 1. Then,*

$$\mathbb{E}_k \left[ \operatorname{grad} f_{B_k}(x_k) \right] = \operatorname{grad} f(x_k),$$

*for all $k \geq 1$.*

*Proof.* From (1), Assumption 3.1 (A1) and the linearity of $\mathbb{E}_k[\cdot]$, we have

$$\mathbb{E}_k \left[ \operatorname{grad} f_{B_k}(x_k) \right] = \frac{1}{b_k} \sum_{i=1}^{b_k} \mathbb{E}_k \left[ \operatorname{grad} f_{s_{k,i}}(x_k) \right] = \operatorname{grad} f(x_k).$$

This completes the proof. □

**Lemma A.2.** *Suppose that Assumptions 3.1 (A1) and (A2) hold. Let $(x_k)_{k=1}^\infty$ be a sequence generated by Algorithm 1. Then,*

$$\mathbb{E}_k \left[ \|\operatorname{grad} f_{B_k}(x_k)\|_2^2 \right] \leq \frac{\sigma^2}{b_k} + \|\operatorname{grad} f(x_k)\|_2^2$$

*for all $k \geq 1$.*

*Proof.* From $\|a+b\|_2^2 = \|a\|_2^2 + 2 \langle a, b \rangle_2 + \|b\|_2^2$, we obtain

$$\mathbb{E}_k \left[ \|\operatorname{grad} f_{B_k}(x_k)\|_2^2 \right] = \mathbb{E}_k \left[ \|\operatorname{grad} f_{B_k}(x_k) - \operatorname{grad} f(x_k)\|_2^2 \right]$$
$$+ 2\mathbb{E}_k \left[ \langle \operatorname{grad} f_{B_k}(x_k) - \operatorname{grad} f(x_k), \operatorname{grad} f(x_k) \rangle_2 \right] + \mathbb{E}_k \left[ \|\operatorname{grad} f(x_k)\|_2^2 \right], \quad (13)$$

for all $k \geq 1$. From (1) and Assumption 3.1 (A1), the first term on the right-hand side of (13) yields

$$\mathbb{E}_k \left[ \|\operatorname{grad} f_{B_k}(x_k) - \operatorname{grad} f(x_k)\|_2^2 \right] = \mathbb{E}_k \left[ \left\| \frac{1}{b_k} \sum_{i=1}^{b_k} \operatorname{grad} f_{s_{k,i}}(x_k) - \operatorname{grad} f(x_k) \right\|_2^2 \right]$$
$$= \frac{1}{b_k^2} \mathbb{E}_k \left[ \sum_{i=1}^{b_k} \|\operatorname{grad} f_{s_{k,i}}(x_k) - \operatorname{grad} f(x_k)\|_2^2 \right]$$
$$\leq \frac{\sigma^2}{b_k},$$

where the second equality comes from Assumption 3.1 (A2). From Lemma A.1, the second term on the right-hand side of (13) yields

$$2\mathbb{E}_k \left[ \langle \operatorname{grad} f_{B_k}(x_k) - \operatorname{grad} f(x_k), \operatorname{grad} f(x_k) \rangle_2 \right] = 2 \langle \mathbb{E}_k[\operatorname{grad} f_{B_k}(x_k)] - \operatorname{grad} f(x_k), \operatorname{grad} f(x_k) \rangle_2$$
$$= 2 \langle \operatorname{grad} f(x_k) - \operatorname{grad} f(x_k), \operatorname{grad} f(x_k) \rangle_2$$
$$= 0.$$

Therefore, we obtain

$$\mathbb{E}_k \left[ \|\operatorname{grad} f_{B_k}(x_k)\|_2^2 \right] \leq \frac{\sigma^2}{b_k} + \|\operatorname{grad} f(x_k)\|_2^2,$$

for all $k \geq 1$. This completes the proof. □

**Lemma A.3.** *Suppose that Assumption 3.1 (A3) holds. Then, the sequence $(x_k)_{k=1}^{\infty} \subset M$ generated by Algorithm 2 satisfies*

$$\hat{v}_{k,i} \leq B^2,$$

*for all $k \geq 1$ and $i = 1, \ldots, d$.*

*Proof.* Note that from Assumption 3.1 (A3), we have

$$g_{k,i}^2 \leq g_{k,1}^2 + \cdots + g_{k,d}^2 = \|g_k\|_2^2 \leq B^2$$

for all $k \geq 1$ and $i = 1, \ldots, d$. The proof is by induction. For $k = 1$, from $0 \leq \beta_2 < 1$, we have

$$\hat{v}_{1,i} = v_{1,i} := \beta_2 v_{0,i} + (1 - \beta_2)g_{1,i}^2 = (1 - \beta_2)g_{1,i}^2 \leq g_{1,i}^2 \leq B^2.$$

Suppose that $\hat{v}_{k-1,i} \leq B^2$. From $v_{k-1,i} \leq \hat{v}_{k-1,i} \leq B^2$, we have

$$v_{k,i} = \beta_2 v_{k-1,i} + (1 - \beta_2)g_{k,i}^2 \leq \beta_2 B^2 + (1 - \beta_2)B^2 = B^2.$$

Thus, induction ensures that $v_{k,i} \leq B^2$ for all $k \geq 1$. $\qquad\square$

# B    Proof of Lemma 3.3

*Proof.* We denote grad $f(x_k)$ by $g(x_k)$. From Proposition 3.2, we have

$$f(x_{k+1}) \leq f(x_k) + \left\langle g(x_k), -\alpha_k P_{x_k}(H_k^{-1} g_k) \right\rangle_2 + \frac{L}{2} \left\| -\alpha_k P_{x_k}(H_k^{-1} g_k) \right\|_2^2,$$

for all $k \geq 1$. Here, we note that the tangent space $T_{x_k} M$ of an embedded submanifold $M$ in Euclidean space is a subspace of Euclidean space. Therefore, according to the result in (Axler, 2024, 6.57 (a)), the projection $P_{x_k}$ is a linear map. From the linearity and symmetry of $P_{x_k}$, we obtain

$$\left\langle g(x_k), -\alpha_k P_{x_k}(H_k^{-1} g_k) \right\rangle_2 = \left\langle P_{x_k}(g(x_k)), -\alpha_k H_k^{-1} g_k \right\rangle_2 = \left\langle g(x_k), -\alpha_k H_k^{-1} g_k \right\rangle_2.$$

From the symmetry of $P_{x_k}$ and $P_{x_k} \circ P_{x_k} = P_{x_k}$, we have

$$\begin{aligned}
\left\| -\alpha_k P_{x_k}(H_k^{-1} g_k) \right\|_2^2 &= \alpha_k^2 \left\| P_{x_k}(H_k^{-1} g_k) \right\|_2^2 \\
&= \alpha_k^2 \left\langle P_{x_k}(H_k^{-1} g_k), P_{x_k}(H_k^{-1} g_k) \right\rangle_2 \\
&= \alpha_k^2 \left\langle H_k^{-1} g_k, P_{x_k}(P_{x_k}(H_k^{-1} g_k)) \right\rangle_2 \\
&= \alpha_k^2 \left\langle H_k^{-1} g_k, P_{x_k}(H_k^{-1} g_k) \right\rangle_2 \\
&\leq \alpha_k^2 \left\| H_k^{-1} g_k \right\|_2 \left\| P_{x_k}(H_k^{-1} g_k) \right\|_2.
\end{aligned}$$

Here, when $P_{x_k}(H_k^{-1} g_k) \neq 0 \in \mathbb{R}^d$, it follows that

$$\left\| -\alpha_k P_{x_k}(H_k^{-1} g_k) \right\|_2^2 \leq \alpha_k^2 \left\| H_k^{-1} g_k \right\|_2^2 \leq \alpha_k^2 \nu^2 \|g_k\|_2^2,$$

where the first inequality comes from $\left\| P_{x_k}(H_k^{-1} g_k) \right\|_2 \leq \left\| H_k^{-1} g_k \right\|_2$ (see Bauschke & Combettes (2011, Corollary 3.24 (ii)) and Axler (2024, 6.57 (h)) for details) and the second inequality comes from $O \prec H_k^{-1} \preceq \nu I_d$. On the other hand, this inequality clearly holds if $P_{x_k}(H_k^{-1} g_k) = 0 \in \mathbb{R}^d$. Therefore, we obtain

$$f(x_{k+1}) \leq f(x_k) + \left\langle g(x_k), -\alpha_k H_k^{-1} g_k \right\rangle_2 + \frac{L \alpha_k^2 \nu^2}{2} \|g_k\|_2^2,$$

for all $k \geq 1$. This completes the proof. $\qquad\square$

## C  Linear algebra lemma

**Lemma C.1.** *Let $a = (a_1, \ldots, a_n)^\top \in \mathbb{R}^n$, $b = (b_1, \ldots, b_n)^\top \in \mathbb{R}^n$ and $D = \mathrm{diag}(d_1, \ldots, d_n) \in \mathcal{S}_+^n \cap \mathcal{D}^n$. If $\|a\|_2 \leq A$ and $\|b\|_2 \leq B$, then*

$$a^\top Db \leq AB \, \mathrm{tr}(D).$$

*Proof.* From $\|a\|_2 \leq A$ and $\|b\|_2 \leq B$, we have $|a_i| \leq A$ and $|b_i| \leq B$ for all $i = 1, \ldots, n$. Therefore, we obtain

$$a^\top Db = \sum_{i=1}^n a_i d_i b_i \leq \sum_{i=1}^n |a_i| \cdot |b_i| \, d_i \leq AB \sum_{i=1}^n d_i \leq AB \, \mathrm{tr}(D).$$

This completes the proof. □

## D  Details of numerical experiments

Tables 3–10 summarize the initial step size, batch size, hyperparameters, and CPU time (seconds) per iteration used in the numerical experiments. Table 3 (resp. Table 4) summarizes the details of solving the PCA problem for the MNIST dataset by using the algorithms with constant (resp. diminishing) step sizes. Table 5 (resp. Table 6) summarizes the details of solving the PCA problem for the COIL100 dataset by using the algorithms with constant (resp. diminishing) step sizes. Table 7 (resp. Table 8) summarizes the details of solving the LRMC problem for the MovieLens-1M dataset by using the algorithms with constant (resp. diminishing) step sizes. Table 9 (resp. Table 10) summarizes the details of solving the LRMC problem for the Jester dataset by using the algorithms with constant (resp. diminishing) step sizes.

Table 3: Summary of the initial step size, batch size, CPU time (seconds) per iteration, and hyperparameters used in the case of constant step sizes for the PCA problem on the MNIST dataset.

|  | $\alpha$ | $\beta_1$ | $\beta_2$ | $\beta$ | $\epsilon$ | $b$ | CPU time per iteration (seconds) |
|---|---|---|---|---|---|---|---|
| RSGD | $10^{-2}$ | - | - | - | - | $2^{10}$ | $1.4 \times 10^{-2}$ |
| RAdam | $10^{-2}$ | 0.9 | 0.999 | - | $10^{-8}$ | $2^{10}$ | $1.481 \times 10^{-2}$ |
| RAMSGrad | $10^{-3}$ | 0.9 | 0.999 | - | $10^{-8}$ | $2^{10}$ | $1.563 \times 10^{-2}$ |
| RASA-L | $10^{-3}$ | - | - | 0.99 | $10^{-8}$ | $2^{10}$ | $2.155 \times 10^{-2}$ |
| RASA-R | $10^{-3}$ | - | - | 0.99 | $10^{-8}$ | $2^{10}$ | $1.836 \times 10^{-2}$ |
| RASA-LR | $10^{-3}$ | - | - | 0.99 | $10^{-8}$ | $2^{10}$ | $2.374 \times 10^{-2}$ |

Table 4: Summary of the initial step size, batch size, CPU time (seconds) per iteration, and hyperparameters used in the case of diminishing step sizes for the PCA problem on the MNIST dataset.

|  | $\alpha$ | $\beta_1$ | $\beta_2$ | $\beta$ | $\epsilon$ | $b$ | CPU time per iteration (seconds) |
|---|---|---|---|---|---|---|---|
| RSGD | $10^{-1}$ | - | - | - | - | $2^{10}$ | $1.19 \times 10^{-2}$ |
| RAdam | $10^{-1}$ | 0.9 | 0.999 | - | $10^{-8}$ | $2^{10}$ | $1.275 \times 10^{-2}$ |
| RAMSGrad | $10^{-2}$ | 0.9 | 0.999 | - | $10^{-8}$ | $2^{10}$ | $1.28 \times 10^{-2}$ |
| RASA-L | $10^{-2}$ | - | - | 0.99 | $10^{-8}$ | $2^{10}$ | $1.803 \times 10^{-2}$ |
| RASA-R | $10^{-2}$ | - | - | 0.99 | $10^{-8}$ | $2^{10}$ | $1.629 \times 10^{-2}$ |
| RASA-LR | $10^{-2}$ | - | - | 0.99 | $10^{-8}$ | $2^{10}$ | $2.222 \times 10^{-2}$ |

## E  Extended results on batch size comparisons

This appendix presents a detailed comparison of using different batch sizes across various datasets and experiments. Tables 11–16 show the performance of the algorithms under varying batch sizes, highlighting the impact on convergence speed and computational cost. The results further support our analyses (Theorems 3.5 and 3.6) that larger batch sizes lead to faster convergence.

Table 5: Summary of the initial step size, batch size, CPU time (seconds) per iteration, and hyperparameters used in the case of constant step sizes for the PCA problem on the COIL100 dataset.

| | $\alpha$ | $\beta_1$ | $\beta_2$ | $\beta$ | $\epsilon$ | $b$ | CPU time per iteration (seconds) |
|---|---|---|---|---|---|---|---|
| RSGD | $10^{-2}$ | - | - | - | - | $2^{10}$ | $3.937 \times 10^{-2}$ |
| RAdam | $10^{-2}$ | 0.9 | 0.999 | - | $10^{-8}$ | $2^{10}$ | $5.256 \times 10^{-2}$ |
| RAMSGrad | $10^{-3}$ | 0.9 | 0.999 | - | $10^{-8}$ | $2^{10}$ | $4.531 \times 10^{-2}$ |
| RASA-L | $10^{-3}$ | - | - | 0.99 | $10^{-8}$ | $2^{10}$ | $5.697 \times 10^{-2}$ |
| RASA-R | $10^{-3}$ | - | - | 0.99 | $10^{-8}$ | $2^{10}$ | $5.13 \times 10^{-2}$ |
| RASA-LR | $10^{-3}$ | - | - | 0.99 | $10^{-8}$ | $2^{10}$ | $5.859 \times 10^{-2}$ |

Table 6: Summary of the initial step size, batch size, CPU time (seconds) per iteration, and hyperparameters used in the case of diminishing step sizes for the PCA problem on the COIL100 dataset.

| | $\alpha$ | $\beta_1$ | $\beta_2$ | $\beta$ | $\epsilon$ | $b$ | CPU time per iteration (seconds) |
|---|---|---|---|---|---|---|---|
| RSGD | $10^{-1}$ | - | - | - | - | $2^{10}$ | $3.638 \times 10^{-2}$ |
| RAdam | $10^{-2}$ | 0.9 | 0.999 | - | $10^{-8}$ | $2^{10}$ | $5.339 \times 10^{-2}$ |
| RAMSGrad | $10^{-3}$ | 0.9 | 0.999 | - | $10^{-8}$ | $2^{10}$ | $4.033 \times 10^{-2}$ |
| RASA-L | $10^{-2}$ | - | - | 0.99 | $10^{-8}$ | $2^{10}$ | $6.805 \times 10^{-2}$ |
| RASA-R | $10^{-2}$ | - | - | 0.99 | $10^{-8}$ | $2^{10}$ | $5.781 \times 10^{-2}$ |
| RASA-LR | $10^{-2}$ | - | - | 0.99 | $10^{-8}$ | $2^{10}$ | $6.938 \times 10^{-2}$ |

Table 7: Summary of the initial step size, batch size, CPU time (seconds) per iteration, and hyperparameters used in the case of constant step sizes for the LRMC problem on the MovieLens-1M dataset.

| | $\alpha$ | $\beta_1$ | $\beta_2$ | $\beta$ | $\epsilon$ | $b$ | CPU time per iteration (seconds) |
|---|---|---|---|---|---|---|---|
| RSGD | $10^{-3}$ | - | - | - | - | $2^8$ | $2.921 \times 10^{-1}$ |
| RAdam | $10^{-3}$ | 0.9 | 0.999 | - | $10^{-8}$ | $2^8$ | $2.813 \times 10^{-1}$ |
| RAMSGrad | $10^{-3}$ | 0.9 | 0.999 | - | $10^{-8}$ | $2^8$ | $2.806 \times 10^{-1}$ |
| RASA-L | $10^{-3}$ | - | - | 0.99 | $10^{-8}$ | $2^8$ | $3.834 \times 10^{-1}$ |
| RASA-R | $10^{-4}$ | - | - | 0.99 | $10^{-8}$ | $2^8$ | $8.512 \times 10^{-2}$ |
| RASA-LR | $10^{-4}$ | - | - | 0.99 | $10^{-8}$ | $2^8$ | $1.285 \times 10^{-2}$ |

Table 8: Summary of the initial step size, batch size, CPU time (seconds) per iteration, and hyperparameters used in the case of diminishing step sizes for the LRMC problem on the MovieLens-1M dataset.

| | $\alpha$ | $\beta_1$ | $\beta_2$ | $\beta$ | $\epsilon$ | $b$ | CPU time per iteration (seconds) |
|---|---|---|---|---|---|---|---|
| RSGD | $10^{-3}$ | - | - | - | - | $2^8$ | $1.931 \times 10^{-1}$ |
| RAdam | $10^{-3}$ | 0.9 | 0.999 | - | $10^{-8}$ | $2^8$ | $1.953 \times 10^{-1}$ |
| RAMSGrad | $10^{-3}$ | 0.9 | 0.999 | - | $10^{-8}$ | $2^8$ | $2.302 \times 10^{-1}$ |
| RASA-L | $10^{-3}$ | - | - | 0.99 | $10^{-8}$ | $2^8$ | $3.11 \times 10^{-1}$ |
| RASA-R | $10^{-4}$ | - | - | 0.99 | $10^{-8}$ | $2^8$ | $1.231 \times 10^{-2}$ |
| RASA-LR | $10^{-4}$ | - | - | 0.99 | $10^{-8}$ | $2^8$ | $1.86 \times 10^{-2}$ |

Table 9: Summary of the initial step size, batch size, CPU time (seconds) per iteration, and hyperparameters used in the case of constant step sizes for the LRMC problem on the Jester dataset.

| | $\alpha$ | $\beta_1$ | $\beta_2$ | $\beta$ | $\epsilon$ | $b$ | CPU time per iteration (seconds) |
|---|---|---|---|---|---|---|---|
| RSGD | $10^{-3}$ | - | - | - | - | $2^8$ | $6.03 \times 10^{-2}$ |
| RAdam | $10^{-3}$ | 0.9 | 0.999 | - | $10^{-8}$ | $2^8$ | $5.973 \times 10^{-2}$ |
| RAMSGrad | $10^{-3}$ | 0.9 | 0.999 | - | $10^{-8}$ | $2^8$ | $5.966 \times 10^{-2}$ |
| RASA-L | $10^{-3}$ | - | - | 0.99 | $10^{-8}$ | $2^8$ | $5.955 \times 10^{-2}$ |
| RASA-R | $10^{-3}$ | - | - | 0.99 | $10^{-8}$ | $2^8$ | $5.88 \times 10^{-2}$ |
| RASA-LR | $10^{-3}$ | - | - | 0.99 | $10^{-8}$ | $2^8$ | $5.958 \times 10^{-2}$ |

Table 10: Summary of the initial step size, batch size, CPU time (seconds) per iteration, and hyperparameters used in the case of diminishing step sizes for the LRMC problem on the Jester dataset.

| | $\alpha$ | $\beta_1$ | $\beta_2$ | $\beta$ | $\epsilon$ | $b$ | CPU time per iteration (seconds) |
|---|---|---|---|---|---|---|---|
| RSGD | $10^{-3}$ | - | - | - | - | $2^8$ | $5.815 \times 10^{-2}$ |
| RAdam | $10^{-3}$ | 0.9 | 0.999 | - | $10^{-8}$ | $2^8$ | $5.836 \times 10^{-2}$ |
| RAMSGrad | $10^{-3}$ | 0.9 | 0.999 | - | $10^{-8}$ | $2^8$ | $5.885 \times 10^{-2}$ |
| RASA-L | $10^{-3}$ | - | - | 0.99 | $10^{-8}$ | $2^8$ | $5.746 \times 10^{-2}$ |
| RASA-R | $10^{-3}$ | - | - | 0.99 | $10^{-8}$ | $2^8$ | $5.853 \times 10^{-2}$ |
| RASA-LR | $10^{-3}$ | - | - | 0.99 | $10^{-8}$ | $2^8$ | $5.829 \times 10^{-2}$ |

Table 11: Iterations and CPU time (seconds) required by each algorithm with a constant step size to reduce the gradient norm below 4 for solving the PCA problem on the COIL100 dataset, across different batch sizes. A "-" indicates cases where the algorithm did not reach the threshold within the maximum allowed 1,000 iterations.

| | batch size $b$ | number of iterations | CPU time (seconds) |
|---|---|---|---|
| RSGD | 256 | - | - |
| | 512 | - | - |
| | 1024 | - | - |
| RAdam | 256 | - | - |
| | 512 | 19 | 0.178 |
| | 1024 | 14 | 0.159 |
| RAMSGrad | 256 | 132 | 1.105 |
| | 512 | 23 | 0.220 |
| | 1024 | 14 | 0.158 |
| RASA-L | 256 | 297 | 3.558 |
| | 512 | 122 | 1.636 |
| | 1024 | 75 | 1.169 |
| RASA-R | 256 | 53 | 0.490 |
| | 512 | 34 | 0.358 |
| | 1024 | 29 | 0.372 |
| RASA-LR | 256 | 797 | 9.741 |
| | 512 | 225 | 2.968 |
| | 1024 | 60 | 0.936 |

Table 12: Iterations and CPU time (seconds) required by each algorithm with a diminishing step size to reduce the gradient norm below 4 for solving the PCA problem on the COIL100 dataset across different batch sizes.

|  | batch size $b$ | number of iterations | CPU time (seconds) |
|---|---|---|---|
| RSGD | 256 | 237 | 1.695 |
|  | 512 | 171 | 1.526 |
|  | 1024 | 132 | 1.562 |
| RAdam | 256 | 21 | 0.160 |
|  | 512 | 17 | 0.153 |
|  | 1024 | 16 | 0.179 |
| RAMSGrad | 256 | 32 | 0.245 |
|  | 512 | 23 | 0.207 |
|  | 1024 | 20 | 0.247 |
| RASA-L | 256 | 94 | 1.082 |
|  | 512 | 52 | 0.670 |
|  | 1024 | 36 | 0.542 |
| RASA-R | 256 | 19 | 0.168 |
|  | 512 | 12 | 0.113 |
|  | 1024 | 9 | 0.123 |
| RASA-LR | 256 | 36 | 0.420 |
|  | 512 | 31 | 0.400 |
|  | 1024 | 27 | 0.411 |

Table 13: Iterations and CPU time (seconds) required by each algorithm with a constant step size to reduce the gradient norm below 3 for solving the LRMC problem on the MovieLens-1M dataset across different batch sizes. "-" indicates cases where the algorithm did not reach the threshold within the maximum allowed (300) iterations.

|  | batch size $b$ | number of iterations | CPU time (seconds) |
|---|---|---|---|
| RSGD | 64 | 23 | 0.143 |
|  | 128 | 20 | 0.249 |
|  | 256 | 21 | 0.481 |
| RAdam | 64 | 45 | 0.292 |
|  | 128 | 26 | 0.312 |
|  | 256 | 16 | 0.365 |
| RAMSGrad | 64 | - | - |
|  | 128 | - | - |
|  | 256 | 155 | 3.717 |
| RASA-L | 64 | 110 | 4.490 |
|  | 128 | 19 | 0.950 |
|  | 256 | 8 | 0.447 |
| RASA-R | 64 | 37 | 0.637 |
|  | 128 | 34 | 0.745 |
|  | 256 | 34 | 1.106 |
| RASA-LR | 64 | 19 | 0.723 |
|  | 128 | 9 | 0.424 |
|  | 256 | 5 | 0.232 |

Table 14: Iterations and CPU time (seconds) required by each algorithm with a diminishing step size to reduce the gradient norm below 3 for solving the LRMC problem on the MovieLens-1M dataset across different batch sizes. "-" indicates cases where the algorithm did not reach the threshold within the maximum allowed (300) iterations.

|  | batch size $b$ | number of iterations | CPU time (seconds) |
|---|---|---|---|
| RSGD | 64 | 108 | 0.691 |
|  | 128 | 97 | 1.172 |
|  | 256 | 110 | 2.595 |
| RAdam | 64 | 15 | 0.093 |
|  | 128 | 13 | 0.149 |
|  | 256 | 11 | 0.238 |
| RAMSGrad | 64 | 50 | 0.322 |
|  | 128 | 32 | 0.385 |
|  | 256 | 22 | 0.503 |
| RASA-L | 64 | 8 | 0.299 |
|  | 128 | 7 | 0.305 |
|  | 256 | 5 | 0.261 |
| RASA-R | 64 | - | - |
|  | 128 | - | - |
|  | 256 | - | - |
| RASA-LR | 64 | 22 | 0.874 |
|  | 128 | 16 | 0.705 |
|  | 256 | 13 | 0.688 |

Table 15: Iterations and CPU time (seconds) required by each algorithm with a constant step size to reduce the gradient norm below 1/4 for solving the LRMC problem on the Jester dataset across different batch sizes. "-" indicates cases where the algorithm did not reach the threshold within the maximum allowed (300) iterations.

|  | batch size $b$ | number of iterations | CPU time (seconds) |
|---|---|---|---|
| RSGD | 64 | 204 | 0.391 |
|  | 128 | 210 | 0.785 |
|  | 256 | 204 | 1.500 |
| RAdam | 64 | 37 | 0.070 |
|  | 128 | 41 | 0.150 |
|  | 256 | 34 | 0.245 |
| RAMSGrad | 64 | - | - |
|  | 128 | 66 | 0.244 |
|  | 256 | 54 | 0.392 |
| RASA-L | 64 | 32 | 0.062 |
|  | 128 | 36 | 0.134 |
|  | 256 | 32 | 0.231 |
| RASA-R | 64 | 38 | 0.072 |
|  | 128 | 37 | 0.137 |
|  | 256 | 34 | 0.244 |
| RASA-LR | 64 | 174 | 0.342 |
|  | 128 | 48 | 0.184 |
|  | 256 | 43 | 0.312 |

Table 16: Iterations and CPU time (seconds) required by each algorithm with a diminishing step size to reduce the gradient norm below $1/2$ for solving the LRMC problem on the Jester dataset across different batch sizes. "-" indicates cases where the algorithm did not reach the threshold within the maximum allowed (300) iterations.

| | batch size $b$ | number of iterations | CPU time (seconds) |
|---|---|---|---|
| RSGD | 64 | - | - |
| | 128 | - | - |
| | 256 | - | - |
| RAdam | 64 | 188 | 0.376 |
| | 128 | 225 | 0.840 |
| | 256 | 167 | 1.232 |
| RAMSGrad | 64 | 21 | 0.039 |
| | 128 | 18 | 0.064 |
| | 256 | 19 | 0.136 |
| RASA-L | 64 | 184 | 0.372 |
| | 128 | 249 | 0.943 |
| | 256 | 178 | 1.325 |
| RASA-R | 64 | 219 | 0.427 |
| | 128 | - | - |
| | 256 | 195 | 1.441 |
| RASA-LR | 64 | 5 | 0.008 |
| | 128 | 4 | 0.012 |
| | 256 | 3 | 0.015 |

## F   Conditions under which Assumption 3.1 (A2) holds

We introduce the assumptions (Assumptions F.1 (B1) and (B2)) under which Assumption 3.1 (A2) from Section 3.1 holds, and demonstrate that Assumption 3.1 (A2) is valid under these assumptions (Lemma F.2). Note that Assumptions F.1 (B1) and (B2) are simply the counterparts of Assumptions 3.1 (A4) and (A5) for $f$, applied to each $f_i$ $(i = 1, \dots, N)$.

**Assumption F.1.** *For all $i = 1, \dots, N$,*

*(B1) There exists a constant $L_i > 0$ such that*

$$|\mathrm{D}(f_i \circ R_x)(\eta)[\eta] - \mathrm{D}f_i(x)[\eta]| \leq L_i \left\| \eta \right\|_2^2,$$

*for all $x \in M$, $\eta \in T_x M$.*

*(B2) $f_i$ is bounded below by $f_\star \in \mathbb{R}$.*

**Lemma F.2.** *Suppose that Assumptions F.1 (B1) and (B2) hold. We assume that a random variable $s_{k,i}$ takes a value from 1 to $N$ following a uniform distribution. Then, the sequence $(x_k)_{k=1}^\infty \subset M$ generated by Algorithm 1 satisfies*

$$\mathbb{E}_k \left[ \left\| \mathrm{grad}\, f_{s_{k,i}}(x_k) - \mathrm{grad}\, f(x_k) \right\|_2^2 \right] \leq \frac{2}{N} \sum_{i=1}^N L_i M_i,$$

*where*

$$M_i := \sup\{f_i(x_k) - f_\star \mid k \geq 1\} \in [0, \infty).$$

*Proof.* From Assumption 3.1 (A5) and Proposition 3.2, we have

$$
\begin{aligned}
f_* &\le f_i\left(R_{x_k}\left(-\frac{1}{L_i}\operatorname{grad} f_i(x_k)\right)\right) \\
&\le f_i(x_k) + \left\langle \operatorname{grad} f_i(x_k), -\frac{1}{L_i}\operatorname{grad} f_i(x_k)\right\rangle_{x_k} + \frac{L_i}{2}\left\|-\frac{1}{L_i}\operatorname{grad} f_i(x_k)\right\|^2_{x_k} \\
&= f_i(x_k) - \frac{1}{L_i}\|\operatorname{grad} f_i(x_k)\|^2_{x_k} + \frac{1}{2L_i}\|\operatorname{grad} f_i(x_k)\|^2_{x_k} \\
&= f_i(x_k) - \frac{1}{2L_i}\|\operatorname{grad} f_i(x_k)\|^2_{x_k}.
\end{aligned}
$$

Therefore, we obtain

$$
\|\operatorname{grad} f_i(x_k)\|^2_{x_k} \le 2L_i(f_i(x_k) - f_\star) \le 2L_i M_i, \tag{14}
$$

where

$$
M_i := \sup\{f_i(x_k) - f_\star \mid k \ge 1\} \in [0, \infty).
$$

From $\|a - b\|^2_2 = \|a\|^2_2 - 2\langle a, b\rangle_2 + \|b\|^2_2$, we obtain

$$
\begin{aligned}
&\mathbb{E}_k\left[\|\operatorname{grad} f_{s_{k,i}}(x_k) - \operatorname{grad} f(x_k)\|^2_2\right] \\
&= \mathbb{E}_k\left[\|\operatorname{grad} f_{s_{k,i}}(x_k)\|^2_2\right] - 2\mathbb{E}_k\left[\langle \operatorname{grad} f_{s_{k,i}}(x_k), \operatorname{grad} f(x_k)\rangle_2\right] + \mathbb{E}_k\left[\|\operatorname{grad} f(x_k)\|^2_2\right] \\
&= \frac{1}{N}\sum_{i=1}^N \|\operatorname{grad} f_i(x_k)\|^2_2 - 2\left\langle \mathbb{E}_k[\operatorname{grad} f_{s_{k,i}}(x_k)], \operatorname{grad} f(x_k)\right\rangle_2 + \|\operatorname{grad} f(x_k)\|^2_2 \\
&= \frac{1}{N}\sum_{i=1}^N \|\operatorname{grad} f_i(x_k)\|^2_2 - \|\operatorname{grad} f(x_k)\|^2_2 \\
&\le \frac{1}{N}\sum_{i=1}^N \|\operatorname{grad} f_i(x_k)\|^2_2.
\end{aligned}
$$

Here, by using (14), it follows that

$$
\mathbb{E}_k\left[\|\operatorname{grad} f_{s_{k,i}}(x_k) - \operatorname{grad} f(x_k)\|^2_2\right] \le \frac{2}{N}\sum_{i=1}^N L_i M_i.
$$

$\square$

## G Experimental results on the PCA and LRMC with increasing batch sizes

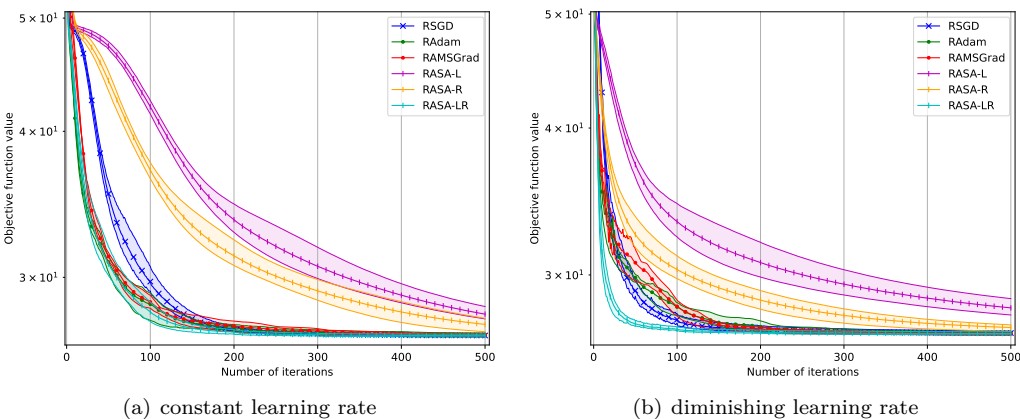

(a) constant learning rate

(b) diminishing learning rate

Figure 17: Objective function value defined by (10) versus number of iterations on the training set of the MNIST datasets.

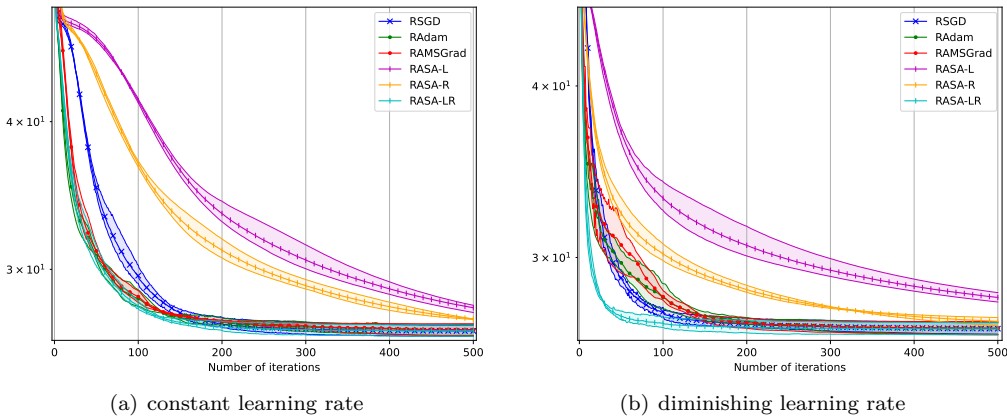

(a) constant learning rate

(b) diminishing learning rate

Figure 18: Objective function value defined by (10) versus number of iterations on the test set of the MNIST datasets.

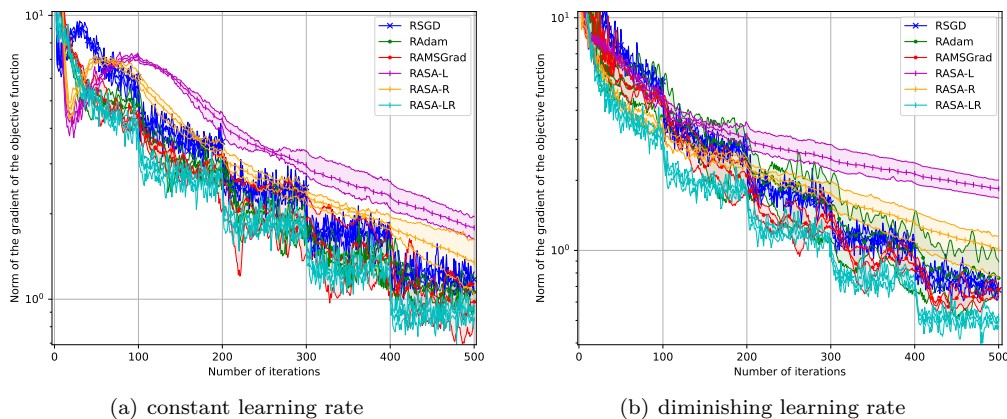

(a) constant learning rate

(b) diminishing learning rate

Figure 19: Norm of the gradient of objective function defined by (10) versus number of iterations on the training set of the MNIST datasets.

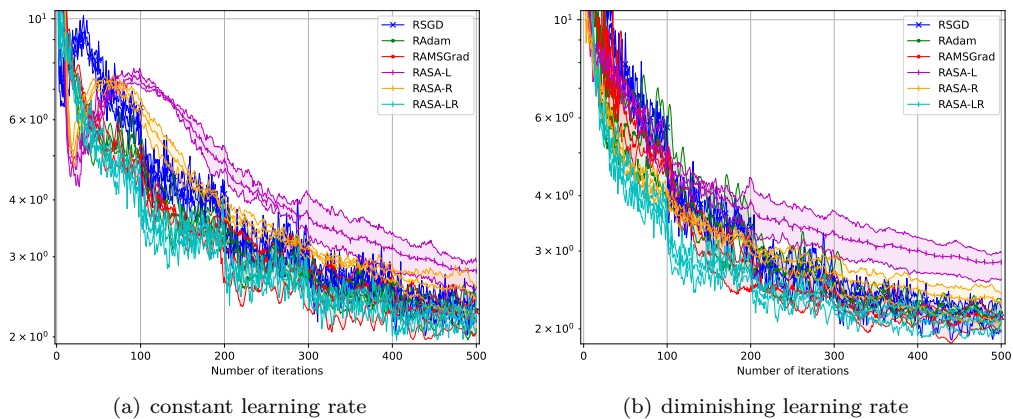

(a) constant learning rate

(b) diminishing learning rate

Figure 20: Norm of the gradient of objective function defined by (10) versus number of iterations on the test set of the MNIST datasets.

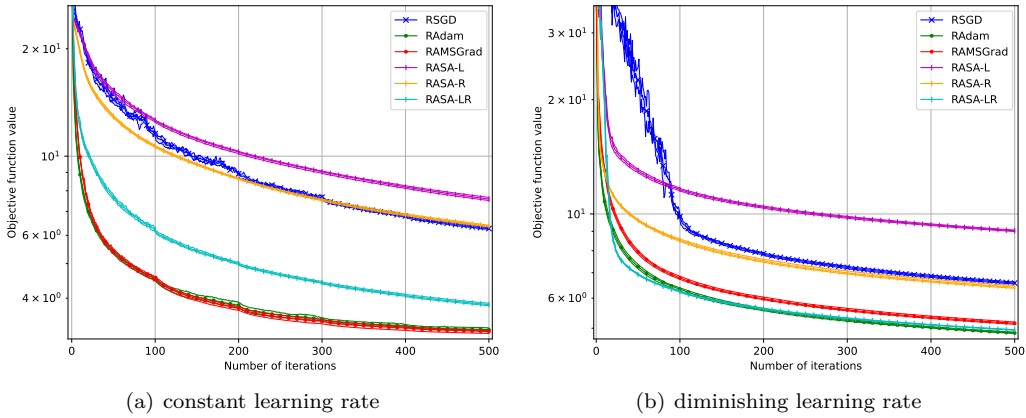

(a) constant learning rate

(b) diminishing learning rate

Figure 21: Objective function value defined by (10) versus number of iterations on the training set of the COIL100 datasets.

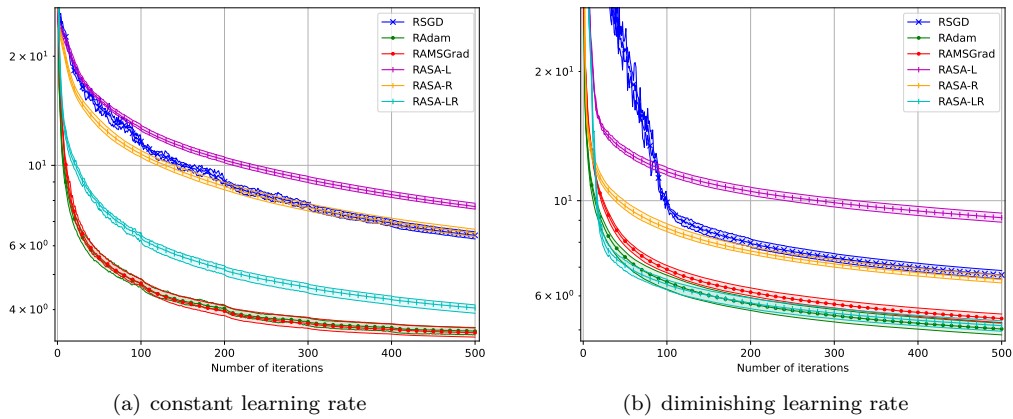

(a) constant learning rate

(b) diminishing learning rate

Figure 22: Objective function value defined by (10) versus number of iterations on the test set of the COIL100 datasets.

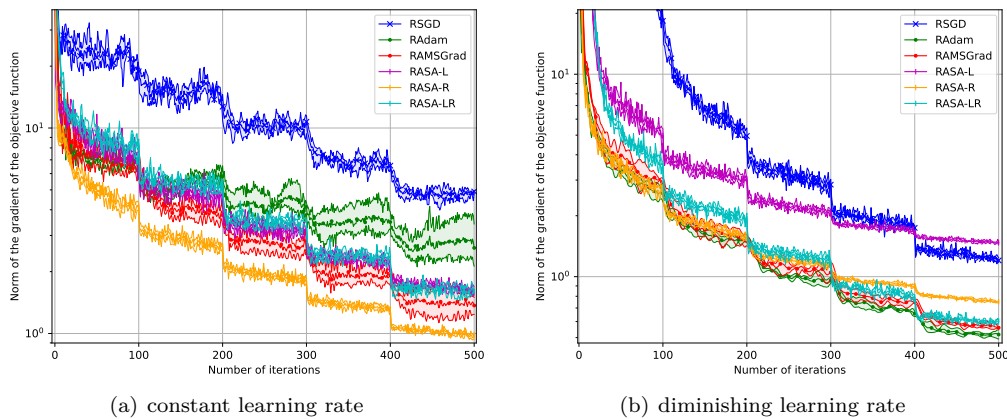

(a) constant learning rate         (b) diminishing learning rate

Figure 23: Norm of the gradient of objective function defined by (10) versus number of iterations on the training set of the COIL100 datasets.

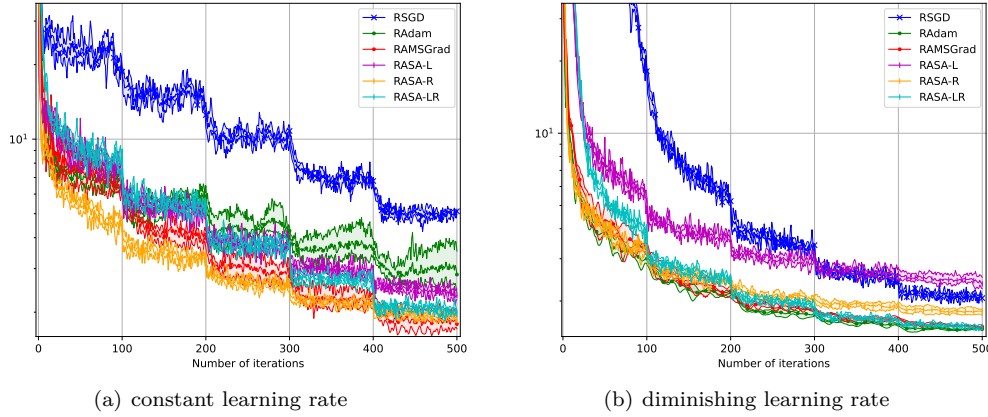

(a) constant learning rate         (b) diminishing learning rate

Figure 24: Norm of the gradient of objective function defined by (10) versus number of iterations on the test set of the COIL100 datasets.

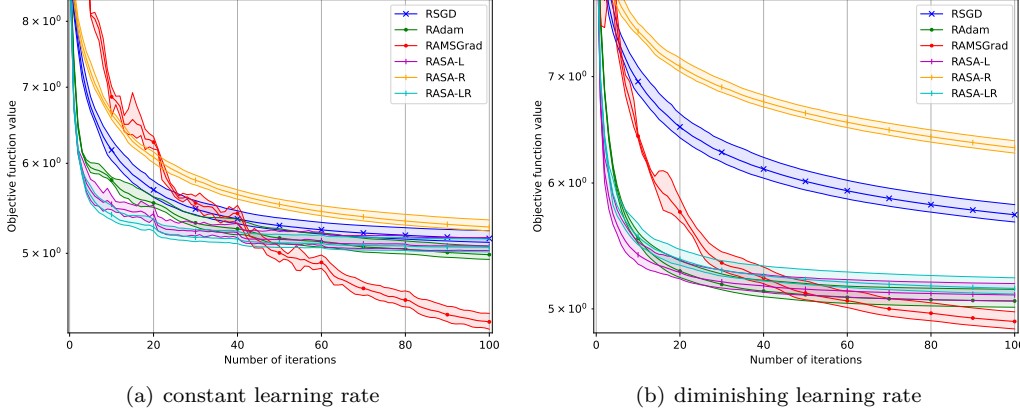

(a) constant learning rate         (b) diminishing learning rate

Figure 25: Objective function value defined by (12) versus number of iterations on the training set of the MovieLens-1M datasets.

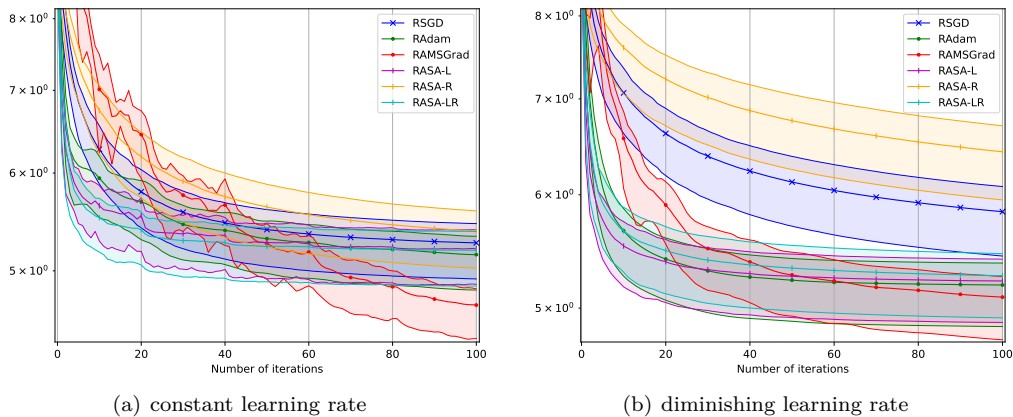

(a) constant learning rate

(b) diminishing learning rate

Figure 26: Objective function value defined by (12) versus number of iterations on the test set of the MovieLens-1M datasets.

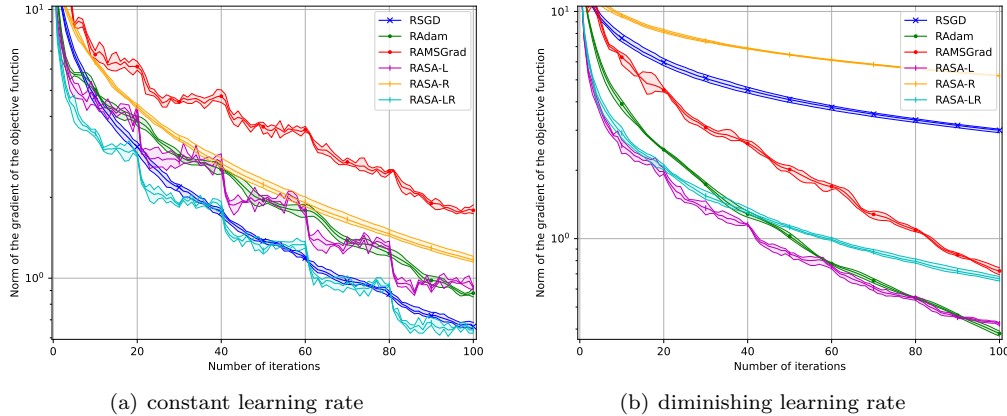

(a) constant learning rate

(b) diminishing learning rate

Figure 27: Norm of the gradient of objective function defined by (12) versus number of iterations on the training set of the MovieLens-1M datasets.

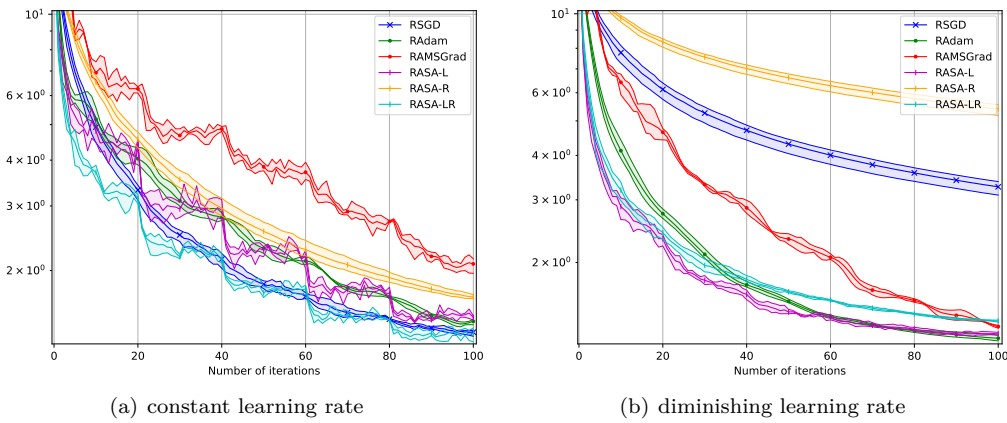

(a) constant learning rate

(b) diminishing learning rate

Figure 28: Norm of the gradient of objective function defined by (12) versus number of iterations on the test set of the MovieLens-1M datasets.

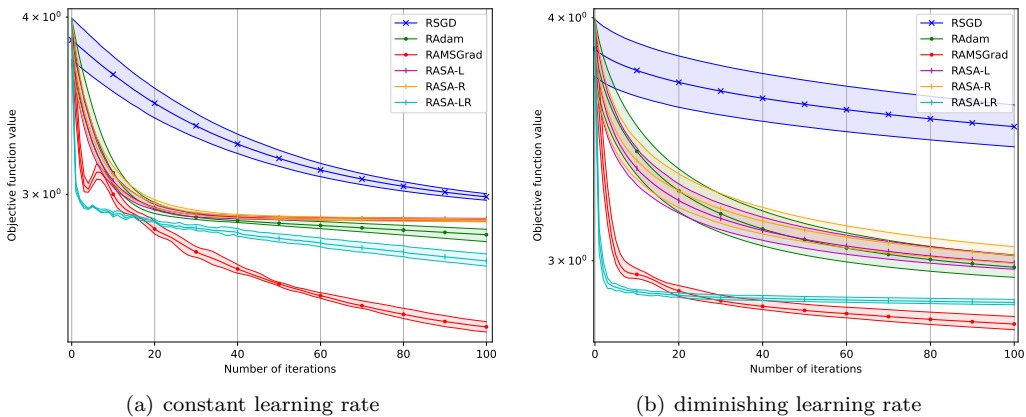

(a) constant learning rate

(b) diminishing learning rate

Figure 29: Objective function value defined by (12) versus number of iterations on the training set of the Jester datasets.

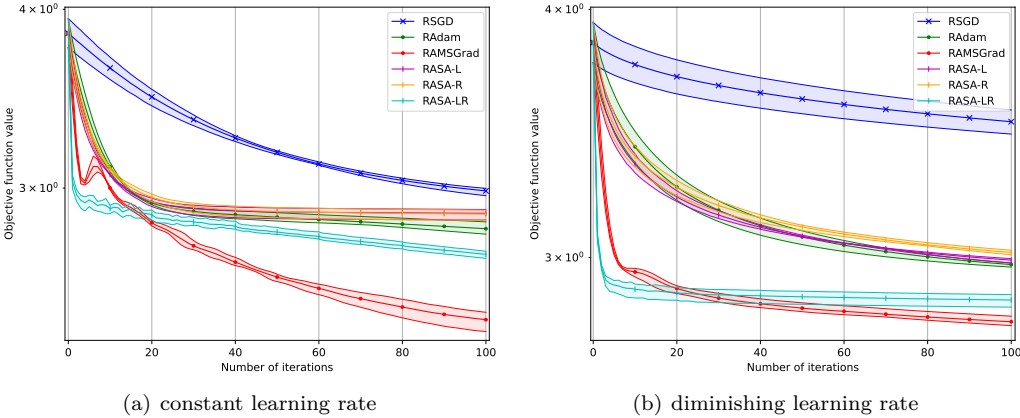

(a) constant learning rate

(b) diminishing learning rate

Figure 30: Objective function value defined by (12) versus number of iterations on the test set of the Jester datasets.

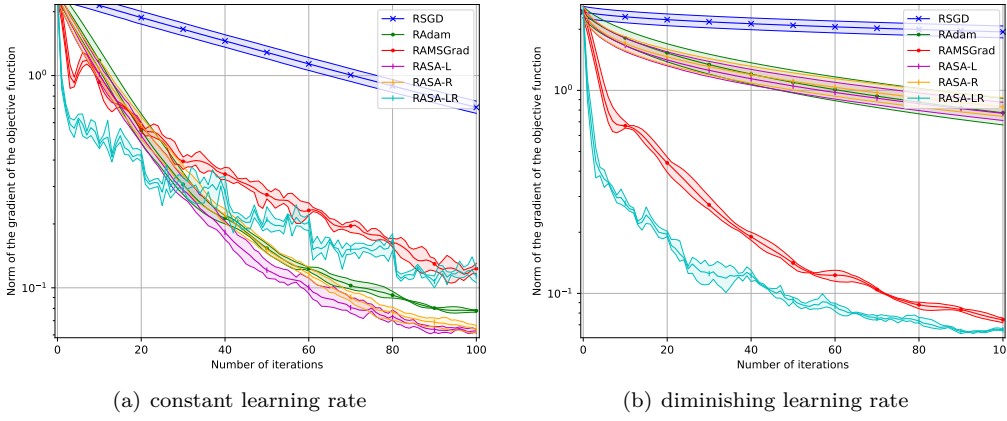

(a) constant learning rate

(b) diminishing learning rate

Figure 31: Norm of the gradient of objective function defined by (12) versus number of iterations on the training set of the Jester datasets.

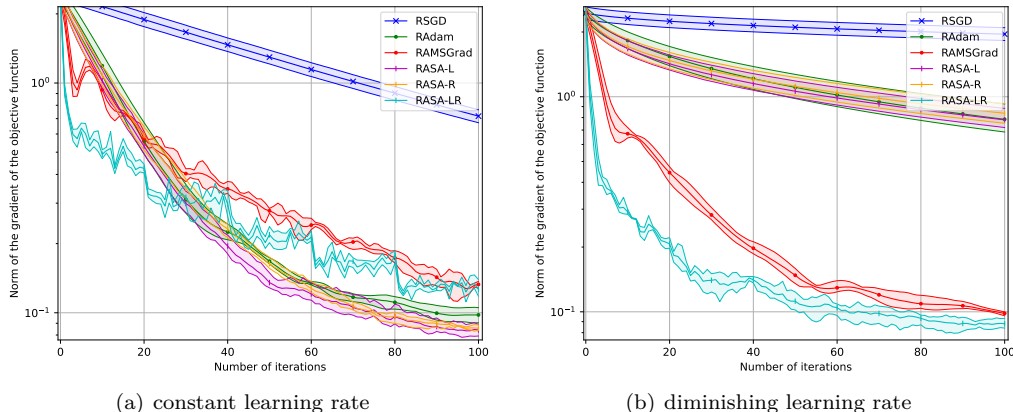

(a) constant learning rate

(b) diminishing learning rate

Figure 32: Norm of the gradient of objective function defined by (12) versus number of iterations on the test set of the Jester datasets.

