# OpenReview forum: "A general framework of Riemannian adaptive optimization methods with a convergence analysis"
_TMLR — Accepted by TMLR_

### Review · Reviewer_jkpK · 2024-10-07

**Summary Of Contributions:**

The paper proposes a general framework for Riemannian adaptive optimization methods, extending well-known stochastic optimization algorithms to Riemannian manifolds. A key contribution is the adaptation of AMSGrad on embedded submanifolds of Euclidean spaces. The authors provide rigorous convergence analyses for both constant and diminishing step sizes, investigating the relationship between convergence rates and mini-batch sizes. The framework is validated on practical tasks like principal component analysis (PCA) and low-rank matrix completion (LRMC). Python implementations are provided, enhancing the reproducibility of the work.

**Audience:**

Yes

**Claims And Evidence:**

Yes

**Requested Changes:**

1. Some sections, particularly the mathematical preliminaries, are dense and difficult to follow. Additional clarifications or simplified explanations could improve accessibility.
2. Displaying the loss on a logarithmic scale in the experiments would make the results clearer and allow for easier comparisons between algorithms.
3. Including test accuracy in the experiments.

### Typos:
Page 2: “unite sphere” -> “unit sphere.”

Page 9: “RAMGRad” ->  “RAMSGrad.”

**Strengths And Weaknesses:**

## Strengths:
1. The manuscript is well-written, and the motivation of the study is clear.
2. The suggested concepts look interesting and promising for Riemannian adaptive optimization methods.
3. The convergence analysis covers both constant and diminishing step sizes, offering a comprehensive theoretical understanding of the proposed framework.
4. The authors demonstrate the framework’s applicability to practical optimization tasks by applying it to real problems such as PCA and LRMC. They provided a GitHub repository for reproducibility.
5. The proof seems clear and correct, and the previous works are well-cited.

## Weaknesses:

While generally well-written, a few changes could improve the clarity of the manuscript. The proposed changes are outlined below and in the requested changes section:

1. Extending the experiments to larger datasets, such as ImageNet or CIFAR-10, would strengthen the empirical validation and demonstrate the scalability of the proposed methods. Why did you choose these datasets?
2. It would be helpful to include in the appendix summary tables for each experiment, detailing the hyperparameters used, such as the learning rate, mini-batch size, and the computation time per epoch. This would improve the clarity of the experimental setup and provide more insights into the computational efficiency of the proposed methods.

---

> ### Author Response · Authors · 2024-10-28
> **Replies to Reviewer jkpK’s comments**
>
> We appreciate your detailed assessments and helpful feedback. We have revised the manuscript to incorporate all of the recommendations, which has resulted in an improved presentation of our work. The revised parts of the manuscript are marked in red.
>
> ### Weaknesses:
> 1. Thank you for your comment. We consider that principal component analysis is more effective for grayscale images than for color images such as CIFAR10 ($3 \times 32 \times 32$ pixels) or ImageNet ($3 \times 224 \times 224$ pixels). Therefore, we chose MNIST ($28\times 28$) and normalized COIL100 ($32\times 32$). In fact, Kasai et al., (2019); Roy et al., (2018) also chose these datasets.
>
> 2. Thank you for your valuable comment. We added the requested tables to Appendix D, detailing the hyperparameters used, such as the learning rate, mini-batch size, and computation time per iteration. We believe these additions will improve the clarity of the experimental setup and provide valuable insights into the computational efficiency of the proposed methods.
>
> ### Requested Changes:
> 1. Thank you for your important comment. In the revision, we have added additional explanations and clarifications to Sections 2 and 3, as well as the Appendix. Furthermore, we included references, specifically to Sakai, (1996); Bauschke & Combettes, (2011) to enhance its understandability and accessibility for readers. We hope these revisions improve the clarity of the mathematical concepts presented.
>
> 2. Thank you for your valuable comment. In the revision, we changed the figures to those of a logarithmic scale.
>
> 3. Thank you for your comment. Since neither of our experiments, i.e., the PCA and LRMC problems, are classification problems, it is not possible to calculate test accuracy. Therefore, we show the objective function value as in previous studies (Kasai et al., 2018; Roy et al., 2018) and plot the norm of the gradient of objective function as a unique contribution of this study. Moreover, in the revision, we split each dataset into a training set and a test set, and plotted the objective function value and norm of the gradient of objective function for the test set as well as the training set.
>
> ### Typos:
> Thank you for your important comment. We have revised the manuscript accordingly.

---

> > ### Comment · Reviewer_jkpK · 2024-10-29
> >
> > Thank you for your answers. Overall, I am satisfied with the answer of the authors.

---

> > > ### Author Response · Authors · 2024-10-30
> > > **Replies to Reviewer jkpK’s official comment**
> > >
> > > We would like to thank you for your careful reading of our manuscript and for the important and helpful comments and suggestions.

---

### Review · Reviewer_RAJx · 2024-10-14

**Summary Of Contributions:**

This paper proposes Riemannian adaptive gradient methods for solving optimization over compact submanifolds. Convergence analysis is presented, and numerical comparisons with existing algorithms are conducted.

**Audience:**

Yes

**Broader Impact Concerns:**

Not found

**Claims And Evidence:**

No

**Requested Changes:**

1. Subsection 1.1. The two paragraphs overlap a lot. Please consider merging them more logically.
2. Line 7 of Section 2. Missing definition of $O$.
3. Line 9 of Section 2.1. Missing definition of tr.
4. Line 11. Should $\eta$ be any vector of $\mathbb{R}^{n\times p}$?
5. Line 12. Missing a comma after i.e.
6. Section 2.2. Is M a submanifold as introduced at the beginning of Section 2? In this case, this would exclude the Grassmann manifold.
7. Second last line of Section 2.2. The local minimizer and the stationary point are distinct concepts in optimization. Why do you need to introduce both?
8. Line 3 of Page 4. What is the definition of $\mathcal{D}^d$?
9. The sentence blow Algorithm 1. I am not sure what is claimed.
10. Lemmas 3.2 and 3.3. These two lemmas are relatively simple and well-understood in literature.  It may not be necessary to include them as separate lemmas.
11. Section 3.1. It is mentioned that averaging between iterations does not work in the Riemannian setting. Since the embedded submanifold is considered and addition is possible, please provide more explanation on why this is a challenge.
12. Lemma 3.5. The result is straightforward and no need to introduce the lemma.
13. Theorem 3.7. The proof appears quite similar to the one in RASA paper. In RASA, a diminishing step size is used to achieve exact convergence instead of convergence to a neighborhood.
14. Theorem 3.9. There doesn’t seem to be much difficulty in establishing the convergence theorem based on the results from the RASA paper. Please clarify the novelty here.
15. Line blow Equation (8). If I understand correctly, the principal component should be in a much smaller dimension of the original data dimension. So, U should be of dimension $p \times r$, where r is the number of principal components.
16. Section 4. Like RASA, the authors should present the convergence curves on both training and testing datasets.
17. Figure 2. The gradient norm is quite large at the end of the iterations. It is hard to draw conclusions.
18. Figure 3. The results on RAdam and RASA seem inconsistent with the RASA paper. Please add explanations.
19. Section 4.2. The analysis is for the submanifold, is it applicable to the Grassmann manifold case?
20. Section 4.2. RASA converges the fastest in the early stage but returns a worse solution than the proposed algorithms at the end. I am wondering whether a smaller (initial) step size would improve the quality of solutions. In addition, please split the training and testing datasets.

**Strengths And Weaknesses:**

Strength: the proposed Riemannian adaptive gradient algorithms is easy to implement by leveraging the ambient Euclidean geometry of the embedded submanifold.

Weakness:
1. The proposed algorithms are not convincingly superior to existing baselines, such as RASA and its variants. A key challenge in designing adaptive gradient methods over Riemannian manifolds is the absence of a coordinate system like that in Euclidean space. The paper would benefit from a clearer explanation of how the proposed methods address this issue.
2. The convergence analysis is limited to a specific setting where no first-order momentum is used. Additionally, the results appear to closely follow the RASA paper, offering no significant advances. The convergence result with constant step size feels more like an intermediate result and does not guarantee obtaining a stationary point in expectation. It seems similar results would also apply to RASA.
3. The writing could be improved, especially in providing intuitions or motivations for the algorithm’s effectiveness. On the theoretical side, the significance of the constant step size and the challenges it presents are not sufficiently explained. Some of the lemmas seem straightforward and could be omitted. In the numerical experiments, it is important to present results on both training and testing datasets for a more comprehensive evaluation.

Based on all these comments, I consider that this paper does not meet the quality requirements for being published in TMLR.

---

> ### Author Response · Authors · 2024-10-28
> **Replies to Reviewer RAJx’s comments**
>
> We appreciate your detailed assessments and helpful feedback. We have revised the manuscript to incorporate all of the recommendations, which has resulted in an improved presentation of our work. The revised parts of the manuscript are marked in red.
>
> ### Weaknesses:
> 1. Thank you for your comment. We agree that the absence of a coordinate system in Riemannian manifolds poses a significant challenge when designing adaptive gradient methods, especially compared to Euclidean space. In this work, we specifically focus on manifolds that are embedded in Euclidean space, a common setting in many applications. We address the issue by constructing algorithms that rely on projections onto the tangent space at each point on the manifold. In the revision, we added the above discussion to the first paragraph of Section 1.2 (contribution section).
>
> 2 and 3. Thank you for your comments. The RASA paper (Kasai et al., 2019) presented useful results indicating RASA with a diminishing step size converges to a stationary point of a continuously differentiable function defined on a matrix manifold. Meanwhile, other previously reported results indicated that using constant step sizes makes the algorithm well used to train deep neural networks (Zaheer et al., 2018) and the setting of the mini-batch size depends on the performance of the algorithms used to train the deep neural networks (Smith et al., 2018; Sato &Iiduka, 2023). The main motivation of this paper is thus to gain an understanding of the theoretical performance relationship between Riemannian adaptive optimization methods using constant step sizes and those using the mini-batch size. We are also interested in understanding the theoretical performance relationship between Riemannian adaptive optimization methods using diminishing step sizes and using the mini-batch size. We added the above discussion to Section 1.1 (Motivation section) of the revision.
>
> The theoretical contribution (the second paragraph of Section 1.2) is to give convergence analyses of the proposed algorithm (Algorithm 1) using a mini-batch size $b$ valid for both a constant step size (Theorem 3.5) and diminishing step size (Theorem 3.6). In particular, Theorem 3.5 indicates that, under some conditions, the sequence $(x_k)\_{k=1}^{\infty}$ generated by the proposed algorithm with a constant step size $\alpha$ satisfies
> \begin{align*}
> \frac{1}{K} \sum\_{k=1}^K \mathbb{E} \left[ \lVert  \mathrm{grad} f(x_k)  \rVert_2^2  \right] = \mathcal{O} \left(\frac{1}{K} + \frac{1}{b}  \right).
> \end{align*}
> Here, Theorem 3.5 is an extension of the result in Zaheer et al., (2018, Corollary 2). Theorem 3.6 indicates that, under some conditions, the sequence $(x_k)\_{k=1}^{\infty}$ generated by the proposed algorithm with a diminishing step size $\alpha_k$ satisfies
> \begin{align*}
> \frac{1}{K} \sum\_{k=1}^K \mathbb{E} \left[ \lVert  \mathrm{grad} f(x_k)  \rVert_2^2  \right] = \mathcal{O} \left( \left(1 + \frac{1}{b}  \right) \frac{\log K}{\sqrt{K}} \right).
> \end{align*}
> Since the stochastic gradient (see (1)) using large mini-batch sizes is approximately the full gradient of $f$, intuitively, using large mini-batch sizes decreases the number of steps needed to find stationary points of $f$. Theorems 3.5 and 3.6 indicate that, the larger the mini-batch size $b$ is, the smaller the upper bound of $\frac{1}{K} \sum\_{k=1}^K \mathbb{E} [ \lVert  \mathrm{grad} f(x_k)  \rVert_2^2]$ becomes and the fewer the required steps become. Hence, Theorems 3.5 and 3.6 would match our intuition.
>
> In the revision, we added the above discussion to Section 1.2 (Contribution section).
>
> The numerical contribution (the third paragraph of Section 1.2) is to show that increasing the batch size decreases the number of steps needed to a certain index, as support for our theoretical analyses. In particular, Table 1 (resp. Table 2) supports Theorem 3.5, demonstrating that the proposed algorithms with a constant (resp. diminishing) step size requires fewer iterations to reduce the gradient norm below the threshold as the batch size increases. Tables 1 and 2 confirm that larger batch sizes lead to faster convergence in terms of number of iterations, aligning with our theoretical analysis. Similar tables (Tables 11-16) for the remaining datasets and experiments can be found in Appendix E.
>
> We added the above discussion to Section 1.2 (Contribution section) of the revision.
>
> In the revision, we moved Lemmas 3.2, 3.3 and 3.5 to the Appendix to simplify the main text. This adjustment should help improve the overall clarity while keeping all necessary details available for reference (please see our responses to Comments 10 and 12 for details). Moreover, we split each dataset into a training set and a test set, and plotted the objective function value and norm of the gradient of the objective function for the test set as well as the training set (please see our responses to Comment 16 for details).

---

> > ### Comment · Reviewer_RAJx · 2024-10-31
> >
> > a. Regarding the mini-batch size, it is well established in the literature that the convergence rate is bounded by \(1/b\), since the noise or variance in the gradients decreases at a rate of 1/b as b increases. The convergence rate with decaying step sizes is similarly straightforward. I have some concerns about the technical novelty and significance.
> >
> > b. For point 11, would it be possible to apply projection after addition to ensure feasibility?

---

> > > ### Author Response · Authors · 2024-11-02
> > > **Replies to Reviewer RAJx’s official comments**
> > >
> > > Thank you for your follow-up comment.
> > >
> > > a. Theorem 3.5 indicates that, although the proposed algorithm with the constant step size $\alpha$ and constant mini-batch size $b$ does not always converge, we can expect that the proposed algorithm with an increasing mini-batch size $b_k$ such that $b\_k \leq b\_{k+1}$ converges. This is because the larger the mini-batch size $b$ is, the fewer the required steps become. In practice, it has been shown that increasing the mini-batch size (Byrd et al., 2012; Balles et al., 2017; De et al., 2017; Smith et al., 2018; Goyal et al., 2018) is useful for training deep neural networks with mini-batch optimizers.
> > > Motivated by Theorems 3.5 and 3.6, we give convergence analyses of the proposed algorithm by using increasing mini-batch size $b\_k$ ($b\_k \leq b\_{k+1}$) valid for both a constant step size (Theorem 3.7) and diminishing step size (Theorem 3.8).
> > > Theorem 3.7 indicates that, under some conditions, the sequence $(x\_k)\_{k=1}^{\infty}$ generated by the proposed algorithm with a constant step size $\alpha$ and an increasing mini-batch size $b_k$ satisfies
> > > \begin{align*}
> > >    \min\_{k=1,\ldots, K} \mathbb{E}\left[\lVert\mathrm{grad} f(x_k)\rVert_2^2\right] \leq \frac{1}{K}\sum\_{k=1}^K\mathbb{E}\left[\lVert\mathrm{grad}f(x_k)\rVert_2^2\right]
> > >    =\mathcal{O}\left(\frac{1}{K}\right),
> > > \end{align*}
> > > that is, the proposed algorithm with increasing mini-batch sizes can find a stationary point of $f$, in contrast to Theorem 3.5. For example, we can set an increasing mini-batch $b_k$ in Theorem 3.7 as doubly increasing mini-batch size.
> > > Moreover, Theorem 3.8 indicates that, under some conditions, the sequence $(x\_k)\_{k=1}^{\infty}$ generated by the proposed algorithm with a diminishing step size $\alpha_k$ and an increasing mini-batch size $b_k$ satisfies
> > > \begin{align*}
> > >    \min\_{k=1,\ldots, K} \mathbb{E}\left[\lVert\mathrm{grad} f(x_k)\rVert_2^2\right] \leq \frac{1}{\sum\_{k=1}^K \alpha_k}\sum_{k=1}^K \alpha_k \mathbb{E}\left[\lVert\mathrm{grad}f(x_k)\rVert_2^2\right]
> > >    =\mathcal{O}\left(
> > >    \frac{1}{\sqrt{K}}\right),
> > > \end{align*}
> > > that is, the proposed algorithm with increasing mini-batch sizes has a better convergence rate than with constant mini-batch sizes shown in Theorem 3.5. The revised parts are marked in blue.
> > > We believe that Theorems 3.7 and 3.8 that guarantee convergence of the algorithm have theoretical contributions, as compared with the literature.
> > >
> > > b. Could you provide more details on what you mean by "applying projection after addition to ensure feasibility".
> > > Knowing the context or specific concerns you had in mind would help us address your question more accurately.
> > > Thank you in advance for any additional explanation you can provide.
> > > If our understanding of your question is correct, we would like to respond as follows:
> > > For any $k$, the projection $P\_{x_k}:\mathbb{R}^d\to T\_{x_k}M$ maps any vector $v$ in Euclidean space $\mathbb{R}^d$ onto the tangent space $T\_{x_k}M$.
> > > Thus, even if we add another vector $w\in\mathbb{R}^d$ before applying the projection, the resulting $P_{x_k}(v+w)$ will still be a tangent vector after projection.

---

> ### Author Response · Authors · 2024-10-28
> **Replies to Reviewer RAJx’s comments**
>
> ### Requested Changes:
> 1. Thank you for your valuable comment. We merged the two paragraphs to describe the contribution of this study in a more concise manner. Please see Subsection 1.2 for details.
>
> 2. Thank you for your important comment. In the revision, we added to Section 2 the sentence “Let $\mathcal{O}$ be Landau's symbol; i.e., for a sequence $(x_k)_{k=0}^\infty$, we write $x_k=\mathcal{O}(g(k))$ as $k\to\infty$ if there exist a constant $C>0$ and an index $k_0$ such that $|x_k|\leq C|g(k)|$ for all $k\geq k_0$. ” in red.
>
> 3. Thank you for your important comment. In the revision, we added to Section 2 the sentence “We denote the trace of a square matrix $A$ by $\mathrm{tr}(A)$.” in red.
>
> 4. Thank you for your comment. Throughout the paper, $\eta$ denotes a tangent vector of the manifold $M$ embedded in $\mathbb{R}^d$. In particular, $\eta$ includes the case where $\eta$ is represented by a vector as well as by a matrix. For example, if we consider a point $x=(1,0)^\top$ in the one-dimensional unit sphere $\mathbb{S}^1=\lbrace x\in\mathbb{R}^2:\lVert x\rVert_2=1\rbrace$, the tangent space at $x$ is represented by $\lbrace(0,t)^\top:t\in\mathbb{R}\rbrace$, so $\eta$ is a vector, i.e., $\eta\in\mathbb{R}^2$.
>
> 5. Thank you for your important comment. We have revised the manuscript accordingly.
>
> 6. Thank you for your comment. While the Grassmann manifold itself is not an embedded submanifold, in practical implementations, computations involving the Grassmann manifold are often performed via the Stiefel manifold. This allows the proposed method to be applied. Such an approach was also used in previous studies, including Kasai et al. (2019); Hu et al. (2024). We added the above discussion to Section 2.1 of the revision.
>
> 7. Thank you for your comment. In the revision, we decided to introduce only the concept of stationary points to reduce confusion.
>
> 8. Thank you for your comment. The definition of $\mathcal{D}^d$ is provided in Section 2, where it is introduced as the set of $d \times d$ diagonal matrices. Please see the first paragraph of Section 2 for details.
>
> 9. Thank you for your valuable comment. In the revision, to clarify our point, we have revised the relevant text to “By setting the maps $(\phi_n)\_{n=1}^\infty$ and $(\psi_n)\_{n=1}^\infty$ in Algorithm 1 to those used by an adaptive optimization method in Euclidean space, any adaptive optimization method can be extended to Riemannian manifolds” in red. Please see Section 2.3 for details.
>
> 10. Thank you for your valuable comment. As you suggested, these lemmas are relatively simple and well-understood in the literature. In the revision, we moved Lemmas 3.2 and 3.3 to the Appendix, where they can still be referenced if necessary. We believe this change enhances the readability of the main paper without sacrificing completeness.

---

> > ### Author Response · Authors · 2024-10-28
> > **Replies to Reviewer RAJx’s comments**
> >
> > 11. Thank you for your comment. We would like to clarify that the embedded submanifold is not closed under addition in Euclidean space. This means that the average of two points on the manifold may not necessarily lie within the manifold itself, which presents a challenge in this context. We appreciate your inquiry, as it prompted us to elaborate on this important distinction. In the revision, we added an explanation to Section 3.2.
> >
> > 12. Thank you for your valuable comment. Similar to Lemmas 3.2 and 3.3, we have moved Lemma 3.5 to the Appendix to simplify the main text. This adjustment should help improve the overall clarity while keeping all necessary details available for reference.
> >
> > 13. Thank you for your comment. Our analysis and the analysis in the RASA paper target entirely different algorithms and manifolds, as we mentioned in our response to Comment 14. Furthermore, we would like to clarify that our analysis can also achieve exact convergence if we employ a diminishing step size, similar to the RASA approach.
> >
> > 14. Thank you for your comment. We would like to clarify that the RASA paper focuses exclusively on the RASA algorithm, which is defined only on matrix manifolds. In contrast, our analysis applies to a broader class of algorithms that can be defined on any manifold embedded in Euclidean space. Therefore, the scope of applicability is fundamentally different. Furthermore, this theorem introduces novelty by establishing a relationship between mini-batches and convergence rates, which was not addressed in the RASA paper. In the revision, we added the additional explanation to Section 1.2 (contribution section).
> >
> > 15. Thank you for your comment. You are correct that the principal components should be in a much smaller dimension than the original data dimension. In our case, $p=10$, which is much smaller than the original data dimension. In the paper, we have indeed followed this approach, where $U$ is of dimension $n\times p$, with $p$ representing the number of principal components.
> >
> > 16. Thank you for your important comment. In the revision, we split each dataset into a training set and a test set, and plotted the objective function value and norm of the gradient of the objective function for the test set as well as the training set.
> >
> > 17. Thank you for your comment. The upper part of the figure is cut off. The initial value of the gradient norm is approximately 38, which is significantly high. In comparison, the gradient norm at the end of the iterations is actually much smaller, indicating that the minimization of the gradient norm was achieved during the optimization process. We added the above discussion to Section 4.1 of the revision.
> >
> > 18. Thank you for your comment. We consider that our results differ from those presented in the RASA paper because of several key factors. In the RASA paper, the mini-batch size was fixed at 10, while in our experiments we used $2^10$. Furthermore, the scope of the grid search differs from that of the RASA paper. These differences in experimental settings are responsible for the observed differences. We added the above discussion to Section 4.1 of the revision.
> >
> > 19. Thank you for your comment. As explained in our response to Comment 6, the analysis in Section 4.2 cannot be directly applied to the Grassmann manifold. However, it can be indirectly applied through the Stiefel manifold. Please see Section 2.1 for details.
> >
> > 20. Thank you for your comment. We would like to clarify that a grid search to determine the initial step size is detailed in Section 4 ($\alpha \in \lbrace 10^{-1}, \ldots, 10^{-8}\rbrace$; Page 10) of the paper. The results indicate that even with a smaller step size, the quality of the solutions is not improved beyond what has already been achieved.

---

> ### Author Response · Authors · 2024-11-17
> **Comment for Reviewer RAJx**
>
> Dear Reviewer RAJx,
>
> We appreciate your detailed assessments and helpful feedback.
> We replied to your two follow-up comments on 02 Nov 2024.
> If you have any questions, please let us know.
>
> We look forward to your valuable comments.
>
> Sincerely yours,
>
> Paper3274 Authors

---

### Review · Reviewer_CFpd · 2024-10-16

**Summary Of Contributions:**

In this article, the authors propose a general framework for stochastic optimization on Riemannian manifolds. They provide in Algorithm 1 a general scheme which, when applied with the right parameters, allows to emcompass previous approaches in the literature. Moreover, they also propose an adaptation of the classical AMSGrad on manifolds in Algorithm 2, and they prove convergence of both algorithms, in the form of decreasing upper bounds on the average of the sequence of expected gradient norms. Finally, they conclude the article with numerical experiments showing that on two different applications (PCA and low-rank matrix completion), using the authors' schemes indeed works well for both decreasing the objective loss functions as well as the gradient norms.

**Audience:**

Yes

**Claims And Evidence:**

Yes

**Requested Changes:**

---Appendix B. There is a square missing in the lhs of the first equation below equation (11).

---Appendix D. There is a square missing in the middle term of the inequalities below "it follows that".

---Appendix D. "comes from $H_k^{-1} \leq \nu I_d$": you should also recall that $H_k^{-1}$ is diagonal with positive coefficients, because otherwise the statement seems false ($B \geq A$ for two general square matrices $A, B$ does not necessarily mean that $\|Bx\| \geq \|Ax\|$).

---Appendix D. "From the linearity and symmetry of $P_{x_k}$": is $P_{x_k}$ really linear? As far as I understand, it is an orthogonal projection onto an affine space, therefore an affine transformation that is not necessarily linear. I also do not see why the norm of $P_{x_k}(y)$ should be less than the norm of $y$ (which is used in the first inequality below "it follows that" with $y = H_k^{-1} g_k$).

**Strengths And Weaknesses:**

The article is well-written and easy to follow, with a particular care in the details of the statements and proofs. This really helps making the work understandable and accessible. The experiments are also well detailed, and easy to reproduce with the code provided by the authors.

As for the weaknesses, I identified two concerns:

---The first is about the list of assumptions in Assumption 3.1. Given that these assumptions are essential to the theoretical results and their proofs, I think it is important to discuss them in more details. This includes, e.g., discussing when should one expect them in practice, what do they mean intuitively, whether they are satisfied or not in different application scenarios, etc. Otherwise, it is a bit difficult to understand whether these are just theoretical statements that almost never occur practically, or if one should expect them to be true in general (without experimental validation).

---The second is about the final message of the experiments. While I appreciate the careful design of both applications, it remains unclear to me what the final message is. I think the authors should elaborate on a few principles that the plots indicate: in what context one optimization method is better or worse than another? How do the experiments help me for choosing the most appropriate optimization method in my own applications? Currently, the analysis of the results is purely comparative, and, as such, it is difficult to figure out what the final message on performance really is.

---

> ### Author Response · Authors · 2024-10-28
> **Replies to Reviewer CFpd’s comments**
>
> We appreciate your detailed assessments and helpful feedback. We have revised the manuscript to incorporate all of the recommendations, which has resulted in an improved presentation of our work. The revised parts of the manuscript are marked in red.
>
> ### Weaknesses:
> 1. Thank you for your comment. Assumptions (A1) and (A2) are satisfied when the probability distribution is a uniform distribution (please see Appendix F) or when the full gradient is used, i.e., when the batch size equals the total number of data. (A1) and (A2) have been used to analyze adaptive methods in Euclidean space (Zaheer et al., 2018, Pages 3, 12, and 16) and on Hadamard manifold (Sakai & Iiduka, 2024, (2.2) and (2.3)). Assumptions (A3)-(A5) are often used in both Euclidean spaces and Riemannian manifolds. Intuitively, in practical applications, these assumptions are satisfied because the search space of the algorithm is bounded. If $f$ is unbounded on an unbounded search space $X$, then (A5) does not hold. Although there is a possibility such that (A4) is satisfied, (A2) would not hold even when the probability distribution is uniform. When $f$ is unbounded, there is no minimizer of $f$. Hence, (A5) is essential to analyze optimization methods. Moreover, (A4) is needed to use a retraction $L$-smooth (Proposition 3.2) that is an extension of the Euclidean-type descent lemma. (A3) holds when the manifold is compact (Absil et al., 2008) or through slight modification of the objective function and the algorithm (Kasai et al., 2018). We added the above discussion to Section 3.1 of the revision.
>
> 2. Thank you for your comment. Based on our experiments, we can summarize the following key findings. For the PCA problem, RAMSGrad with a constant step size minimizes the objective function in fewer or closer iterations compared with RSGD, RASA-L, RASA-R and RASA-LR, regardless of the dataset. For the LRMC problem, RAdam with a diminishing step size minimizes the objective function in fewer or almost the same number of iterations compared with RSGD, RASA-L and RASA-R, irrespective of the dataset. However, for the LRMC problem, RAMSGrad may underperform relative to RASA depending on the dataset. These insights provide practical guidance on selecting the most suitable optimization method based on the specific problem and dataset characteristics. In the revision, we added the above discussion to Section 1.2 (contribution section).
>
> ### Requested Changes:
> 1. Thank you for your important comment. We have revised the manuscript accordingly.
>
> 2. Thank you for your important comment. We have revised the manuscript accordingly.
>
> 3. Thank you for your important comment. In the revision, we replaced $H_k^{-1}\preceq\nu I_d$ with $O\prec H_k^{-1}\preceq\nu I_d$.
>
> 4. Thank you for your comment. The tangent space $T_{x_k}M$ of an embedded submanifold $M$ in Euclidean space is a subspace of Euclidean space, i.e., a closed convex set. Therefore, according to the result in Bauschke & Combettes, (2011, Corollary 3.22 (ii)), the projection $P_{x_k}$ is a linear map. Moreover, from Bauschke & Combettes (2011, Corollary 3.24 (ii)), $\lVert P_{x_k}(y)\rVert_2^2=P_{x_k}(y)^\top y$ holds for any vector $y\in\mathbb{R}^d$. Combining this with the Cauchy-Schwarz inequality, we can further deduce that $\lVert P_{x_k}(y)\rVert_2\leq\lVert y\rVert_2$ also holds. In the revision, we added further explanations on this point to Appendix B.

---

> > ### Comment · Reviewer_CFpd · 2024-10-29
> >
> > Thank you for your answers and explanations.
> >
> > I'm still having trouble with the last comment. As far as I understand, Bauschke & Combettes, (2011, Corollary 3.22 (ii)) only proves that $P_{x_k}$ is affine (and thus not necessarily linear). Similarly, Bauschke & Combettes (2011, Corollary 3.24 (ii)) requires the subspace to be linear, and there is no reason to believe that $T_{x_k}M$ is linear (when it is seen as a subspace of $\mathbb{R}^d$), except when $x_k=0$. Am I missing something?

---

> ### Author Response · Authors · 2024-10-30
> **Replies to Reviewer CFpd’s official comment**
>
> Thank you for your follow-up comment.
> Since $T_{x_k}M$ is a subset of $\mathbb{R}^d$ and, furthermore, is itself a vector space, $T_{x_k}M$ is therefore linear. Therefore, according to the result in Axler (2024, 6.57 (a)) [1] , the projection $P_{x_k}$ onto $T_{x_k} M$ is a linear map.
> We revised the manuscript accordingly. Please see also Page 24 in the revised manuscript.
>
> [1] https://link.springer.com/book/10.1007/978-3-031-41026-0

---

> > ### Comment · Reviewer_CFpd · 2024-10-30
> >
> > Sorry, I still don't see it. In Axler (2024, 6.57 (a)), the result is stated for subspaces, which must contain 0 (see 1.34 in the book). I don't see why $T_{x_k}M$ contains 0.
> >
> > I feel like we are discussing about two different things right now. I made a [drawing](https://drive.google.com/file/d/1qx-STJxXRgYjfm4zNTdUBJXn3ReUXZYo/view) that shows my confusion: in the drawing, the manifold is the sphere embedded in $\mathbb{R}^3$, $x_k$ is at the top, and the tangent plane is in red. Then it is clear that if I take a point $y$ close to the origin, its projection $P_{x_k}(y)$ will have a bigger norm.

---

> > > ### Author Response · Authors · 2024-10-30
> > > **Replies to Reviewer CFpd’s official comment**
> > >
> > > Thank you for your comment. Since $T\_{x_k}M$ is a vector space, it contains 0.
> > >
> > > Important point to resolve your confusion is that Euclidean space $E^\prime$, which contains the tangent space $T\_{x_k}M$ of $x_k$ that is a subspace of $E^\prime$, should be distinguished from Euclidean space $E$ in which the manifold is embedded.
> > > In the case of your drawing, even when considering the tangent space, you have used the origin of Euclidean space $E$.
> > > However, we should consider the Euclidean space $E^\prime$.
> > >
> > > For example, we consider $\mathbb{S}^1=\lbrace x\in\mathbb{R}^2\mid \lVert x\rVert\_2^2=1 \rbrace \subset \mathbb{R}^2$.
> > > The tangent space at $x=(0,1)^\top\in\mathbb{S}^1$ is written by $T\_x\mathbb{S}^1=\lbrace (t,0)\in\mathbb{R}^2\mid t\in \mathbb{R}\rbrace\subset\mathbb{R}^2$.
> > > Here, we note again that if we represent the “$\subset$” for $\mathbb{S}^1$ and the “$\subset$” for $T\_x\mathbb{S}^1$ in a single drawing, the origin is shifted, i.e., the difference between $E=\mathbb{R}^2$ and $E^\prime=\mathbb{R}^2$.
> > > In this case, we consider the tangent vector $\eta=(1,0)^\top\in T\_x\mathbb{S}^1$.
> > > This tangent vector $\eta$ is represented by $(1,1)^\top$ in Euclidean space $E$ in which $\mathbb{S}^1$ is embedded.
> > > However, the correct representation is $\eta=(1,0)^\top$.
> > > In other words, Euclidean space $E^\prime$ whose origin is at the position $x=(0,1)^\top$ is the one that contains $T\_x\mathbb{S}^1$ as a subspace.

---

> > > > ### Comment · Reviewer_CFpd · 2024-10-31
> > > >
> > > > I see, so I guess one can also see $P_{x_k}$ as the projection onto $T_{x_k}M$, after it has been shifted so that it contains $0$. Thank you for your explanations.

---

> > > > > ### Author Response · Authors · 2024-10-31
> > > > > **Replies to Reviewer CFpd’s official comment**
> > > > >
> > > > > We would like to thank you for your careful reading of our manuscript and for the important and helpful comments and suggestions.

---

### Decision · Action_Editor_L31n · 2024-12-09

**Recommendation:** Accept with minor revision

**Comment:**

As mentioned above, I find that there is sufficient novelty to the paper to be of interest to at least some of the TMLR audience. Since there have been no concerns about *correctness* of the results, therefore I'm pleased to find that the paper meets the TMLR criteria.

I have recommended the paper be Accepted with Minor Revisions so that the authors have an opportunity to expand on the conclusions from the numerical experiment.  In particular, a paragraph has been added in the Contributions that expands on this, but this could be further refined:

- the first part of the paragraph reports facts without analysis, e.g.., "RAMSGrad with a constant step size minimizes the objective
function in fewer or closer iterations compared with RSGD, RASA-L, RASA-R and RASA-LR, regardless
of the dataset".  Please add some insight into why this is
- referring to "AMSGrad" is a little confusing, since it's not clear if you're referring to the original algorithm from the literature, or if you refer to your extension. Adding "AMSGrad/Algorithm 2" would be more clear
- the very end of the paragraph does contain some high-level message, mainly that the algorithm converges faster with a larger batch size, but this is only a common-sense sanity check, as it's obvious.  It would instead be interesting to also say something about what the optimal batch size choice is, when taking into account both the work per iteration and the number of iterations.  The actual computation/work depends on implementation, and the convergence rates are only bounds, so this quantity is harder to analyze theoretically... hence it's a good question to answer *experimentally*.

Also, while this paragraph was added in the "Contributions" section, I didn't notice similar high-level commentary in the actual Numerics section. I'd suggest putting some high-level commentary there as well.

(Also, in the revision you can change the URL links as they no longer need to be anonymized)

**Audience:**

I find that at least some of TMLR's audience would be interested.  One reviewer raised concerns over the technical novelty but the other reviewers did not. Looking at the paper myself, I find that the technical novelty rises above the TMLR threshold. This is a timely subject, with new results (e.g., analysis of a RASA-like algorithm but with constant stepsize) that are of interest.

**Claims And Evidence:**

The claims are about convergence (usually about the min or sum of expected norms of gradients, as is common for stochastic methods on non-convex problems), and are backed up by rigorous proofs. No reviewers raised any significant concerns about the proofs.  There are also numerical experiments.

(As a side-note, one reviewer asked for the paper to make it more clear in the text *why* the numerical experiments are run.  I agree; while the text has been updated with small comments, it would be nice to have a paragraph or two of high-level conclusions from this section. It's not easy to find the key take-away without reading the entire section in detail.  The authors have a chance for small revision before the camera-ready submission, so I encourage them to add a paragraph or two about this).

---

> ### Author Response · Authors · 2025-01-09
> **Replies to Action Editor’s official comment**
>
> We would like to extend our heartfelt thanks to your exceptional guidance throughout the review process.
> We have revised the manuscript to incorporate all of the recommendations.
>
> **Comment 1**: Thank you for your comment. As reported in Sakai & Iiduka, 2021, RAMSGrad demonstrates robust performance regardless of the initial step size. For the PCA and LRMC problems (where LRMC adopts a diminishing step size), a grid search was conducted, allowing the selection of an initial step size that yielded even better results. This likely contributed to the strong performance of RAMSGrad across datasets.
>
> However, it is worth noting that the objective function in the LRMC problem is complex, and assumptions such as retraction $L$-smoothness may not hold in this case.
> This could explain the less favorable results observed for RAMSGrad in this particular setting.
>
> We added the above discussion to the final paragraph of Section 1.2 (Contributions section) of the revision.
>
> **Comment 2**: Thank you for your valuable comment. We have revised the relevant sections of the paper as suggested to clarify the distinction by referring to it as "AMSGrad (Algorithm 2)." Please see Section 5 for details.
>
> **Comment 3**: Thank you for your insightful comment. We agree that identifying the optimal batch size by balancing the work per iteration and the number of iterations is a valuable consideration. This idea is closely related to the concept of the critical batch size, which has been discussed even in the context of Riemannian optimization in Sakai & Iiduka, 2024. The existence of a batch size that minimizes stochastic first-order oracle (SFO) calls can theoretically be argued; however, practical determination of this optimal batch size is challenging due to the dependency on unknown parameters such as the gradient Lipschitz constant and the variance of the stochastic gradient.
>
> To address this challenge, an effective approach may involve dynamically increasing the batch size at regular intervals during optimization, such as doubling or tripling it after a fixed number of iterations. This method has theoretical guarantees for convergence (please see Theorems 3.7 and 3.8) and can adaptively approach an efficient balance between computational cost and convergence speed.
> We believe this strategy provides a practical way to address the trade-off.
>
> We added the above discussion to the final paragraph of Section 1.2 (Contributions section) of the revision.